# Deficiency of the HGF/Met pathway leads to thyroid dysgenesis by impeding late thyroid expansion

Ya Fang [1,2,5], Jia-Ping Wan[1,3,5], Zheng Wang[1,5], Shi-Yang Song[1,4,5], Cao-Xu Zhang[1], Liu Yang[1], Qian-Yue Zhang [1], Chen-Yan Yan[1], Feng-Yao Wu[1], Sang-Yu Lu[1], Feng Sun[1], Bing Han[1], Shuang-Xia Zhao [1] ✉, Mei Dong [1] ✉ & Huai-Dong Song [1] ✉

The mechanisms of bifurcation, a key step in thyroid development, are largely unknown. Here we find three zebrafish lines from a forward genetic screening with similar thyroid dysgenesis phenotypes and identify a stop-gain mutation in *hgfa* and two missense mutations in *met* by positional cloning from these zebrafish lines. The elongation of the thyroid primordium along the pharyngeal midline was dramatically disrupted in these zebrafish lines carrying a mutation in *hgfa* or *met*. Further studies show that MAPK inhibitor U0126 could mimic thyroid dysgenesis in zebrafish, and the phenotypes are rescued by overexpression of constitutively active MEK or Snail, downstream molecules of the HGF/Met pathway, in thyrocytes. Moreover, HGF promotes thyrocyte migration, which is probably mediated by downregulation of *E-cadherin* expression. The delayed bifurcation of the thyroid primordium is also observed in thyroid-specific Met knockout mice. Together, our findings reveal that HGF/Met is indispensable for the bifurcation of the thyroid primordium during thyroid development mediated by downregulation of *E-cadherin* in thyrocytes via MAPK-snail pathway.

In all vertebrates, the thyroid gland secretes thyroid hormones, which play multiple roles during the growth and morphogenesis of various organs[1–3] and influence several physiological mechanisms, such as metabolism, reproduction and tissue repair[4]. In mammals, thyroid development is a complex process that includes endodermal precursor cell specification, thyroid bud formation and evagination from the ventral foregut endoderm, thyroid primordium migration to the final pretracheal position, bifurcation of the midline thyroid primordium, formation of the left and right thyroid lobes, functional differentiation of thyroid follicular cells, and the formation of functionally mature thyroid follicles[5,6]. The developmental timing and anatomical organization of the thyroid tissue differs between zebrafish and mammals. In zebrafish, the foregut endoderm cells were specified to have a thyroid fate and were first assembled as a placode in the pharyngeal floor at 26–36 hpf. The thyroid primordium growing along the midline buds off from the pharyngeal floor and moves caudally along the anterior neck region. At 55 hpf, the polarization of the first single thyroid follicular structure takes place in embryos[7]. Then, the thyrocytes detach from the first follicle and migrate along a pair of hypobranchial arteries to form a row of posterior follicles[8], in which T4

[1]Department of Molecular Diagnostics & Endocrinology, The Core Laboratory in Medical Center of Clinical Research, Shanghai Ninth People's Hospital, State Key Laboratory of Medical Genomics, Shanghai Jiao Tong University School of Medicine, Shanghai 200011, China. [2]Department of Endocrinology and Metabolism, The Fourth Affiliated Hospital of Soochow University, Medical Center of Soochow University, Suzhou, Jiangsu 215000, China. [3]Department of Endocrinology, Sir Run Run Shaw Hospital, Zhejiang University School of Medicine, Hangzhou, China. [4]Department of Immunology and Microbiology, Shanghai Institute of Immunology, Shanghai Jiao Tong University School of Medicine, Shanghai, China. [5]These authors contributed equally: Ya Fang, Jia-Ping Wan, Zheng Wang, Shi-Yang Song. ✉e-mail: zhaozhao1215@126.com; dm20180301@163.com; huaidong_s1966@163.com

appears at approximately 72 hpf[7]. Despite the anatomic differences, the molecular mechanisms of thyroid organogenesis appear to be well conserved between zebrafish and mammalian models[9].

Following the fundamental steps of embryonic development, the functions of intrinsic factors, such as the thyroid-specific transcription factor *NKX2.1* (*nkx2.4b* in zebrafish), and *PAX8* (*pax2a* in zebrafish), as well as *FOXE1*, *HHEX*, and *GLIS3*, and extrinsic signaling cues, such as morphogens and growth factors, are integrated to regulate thyroid organogenesis in mice[10]. While the central aspects of thyroid development have been extensively studied, many questions regarding the normal regulation of thyroid development remain unanswered. For example, sonic hedgehog (Shh) was found to regulate the transverse elongation of the thyroid primordium along the third pharyngeal arch arteries after its descent[6,11-13]. However, Shh is not expressed in or close to the migrating primordium; this effect is likely secondary to severe malformations of the vascular tree emerging from the outflow tract[14]. The intrinsic factors responsible for elongation of thyroid development along the third pharyngeal arch arteries (bifurcation) have not been identified. In humans, thyroid dysgenesis (TD) accounts for the majority of congenital hypothyroidism (CH) cases. However, fewer than 10% of TD cases can be identified on the basis of known candidate genes related to thyroid development[15]. Therefore, the underlying molecular mechanism for TD remains largely unknown[16]. This is, in part, due to our limited knowledge of the morphogenetic events underlying thyroid organogenesis[4,17]. Thus, a more sophisticated understanding of the molecular mechanisms mediating thyroid development would be helpful for elucidating the pathogenesis of TD.

In this work, we obtain three zebrafish lines—SH-zH010036 (Shanghai-Zebrafish Hypothyroidism line 010036, abbreviated 010036), SH-zH034060 (034060) and SH-zH018016 (018016)—through forward genetic screening with N-ethyl-N-nitrosourea (ENU) for the regulators of thyroid development in our laboratory[18]. In these three lines, the zebrafish with TD were characterized by a globular appearance of the thyroid gland. Positional cloning was used to identify mutations in hgfa[K80X], met[I284N], and met[E217K] from lines 010086, 034060 and 010036, respectively. Hepatocyte growth factor alpha (hgfa), produced by stromal and mesenchymal cells, is a member of the plasminogen-related growth factor family and stimulates epithelial cell proliferation, motility, morphogenesis and angiogenesis in various organs via tyrosine phosphorylation of its cognate receptor, Met[19]. Notably, compared with the 48 to 120 hpf wild-type embryos, the caudal elongation of thyroid primordia along the pharyngeal midline is disrupted in the embryos of the three zebrafish mutants hgfa[K80X], met[I284N], and met[E217K], leading to the deficiency of the mature thyroid follicles except for the most anterior follicle; the latter was formed at the position of the aortic arch artery 1 bifurcation at 55 hpf[20]. Using a zebrafish model and cell line for in vivo and in vitro studies, we found that HGF/Met signaling plays an indispensable role in regulating the late expansion of thyroid primordia along pharyngeal vessels during thyroid development through the downregulation of E-cadherin expression in thyrocytes via the MAPK pathway. This concept is further supported by the impeded bifurcation of the thyroid primordia in thyroid-specific Met knockout (Met-CKO) mice.

## Results

### Identification of thyroid dysgenesis lines

To discover genes essential for thyroid morphogenesis and development, we conducted ENU-mediated genetic screening in zebrafish using the in situ expression patterns of *tg* (thyroglobulin) and *tshbα* (thyrotrophin) as readouts. A total of 1606 F2 lines were screened, among which 112 F2 mutant lines with normal development stages except for thyroid dysfunction were identified[18]. We observed three lines (designated 010036, 034060 and 018016) with similar phenotypes: in approximately 25% of the embryos, the thyroid appeared grossly as a compact and coherent group of cells and looked much

shorter than that in other siblings in the same litter at 5 dpf (Fig. 1A–C). Because the phenotypes of these three lines were similar, we carried out cross-complementary experiments among these three lines to rule out the possibility that they were genetically the same mutants. Among the offspring of 010036 crossed with 034060, the abnormal small thyroid phenotype was recurrent, but among the offspring of 010036 crossed with 018016 or 034060 crossed with 018016, there was no abnormal phenotype, indicating that the molecular underpinnings of the 010036 and 034060 phenotypes were the same.

### Identification of the pathogenic genes by positional cloning

We performed whole-exome sequencing for the cases (20 affected embryos pooled together) and controls (20 nonaffected sibling embryos pooled together) to identify the molecular impairment responsible for the abnormal thyroid phenotype in the above lines by bulk segregant analysis (BSA)[21] combined with fine positional cloning. Linkage analysis via the BSA method indicated that the potential pathogenic gene was probably located on chromosome 4 in the 018016 line (Supplementary Fig. 1A) and on chromosome 25 in both the 034060 and 010036 lines (Supplementary Fig. 1B, C). After filtering the zebrafish dbSNP database and whole-exome sequencing data produced from more than 600 embryos in 15 zebrafish lines without the TD phenotype at our sequencing center, 25 homozygous variants in 16 genes on chromosome 4 were identified from the affected embryo samples of the 018016 line (Supplementary Table 1). Notably, a stop-gain variant in the coding sequence of the *hgfa* gene (p.Lys80*, Supplementary Table 1) was identified from the 018016 line. We further found that the mutation in *hgfa* was perfectly cosegregated with the 100 embryos with TD of line 018016; however, the wild-type alleles of *ATP2B1A* and *BX324003.3*, which are located upstream and downstream of *hgfa*, respectively, were identified in the affected embryos of line 018016 (Supplementary Table 1), indicating that the mutation of *hgfa* was probably the pathogenic gene in line 018016. Interestingly, only one gene on chromosome 25 with a homozygous mutation in the affected embryos was identified in both the 010036 and 034060 lines (p.Glu217Lys and p.Ile284Asn, respectively; Supplementary Tables 2 and 3). Because the 010036 and 034060 lines carried the same pathogenic gene, as confirmed by cross-complementary experiments, it is tempting to presume that *met* was the causal gene in both lines. We further verified these mutations in the embryos of the mutants and siblings by using specific primers (Fig. 1D–F). We then renamed the three lines hgfa[K80X], met[E217K] and met[I284N] according to the mutation. Interestingly, Met is the receptor for Hgf encoded by the *hgfa* gene in zebrafish (Fig. 1G). The total length of hgfa is 712 amino acids, and the p.K80X mutation in *hgfa* from the hgfa[K80X] line is predicted to produce a truncated form lacking almost all domains (Fig. 1H). To further confirm that *hgfa* or *met* deficiency could lead to a TD phenotype in zebrafish, we injected embryos with translation-blocking *hgfa* morpholinos (MO-*hgfa*) and *met* morpholinos (MO-*met*). We found that both of them could phenocopy the TD phenotype in 5 dpf larvae by whole-mount in situ hybridization (WISH) with a *tg* probe (Fig. 1I, J).

To clarify the role of HGF signaling in thyroid morphogenesis, we examined the expression of *hgfa* and *met* during several stages of thyroid development. At 48 hpf, the thyroid primordium had formed, and *hgfa* was expressed adjacent to the thyroid bud (Fig. 2Aa′, arrowhead). As the thyroid developed, the expression of *hgfa* decreased because it remained low at 54 hpf (Fig. 2Ab′, arrowhead) and was difficult to detect at 60 hpf by WISH. However, the *met* mRNA was colocalized with the *tg* protein in thyrocytes during 48–60 hpf according to WISH (Fig. 2Ac′–Ae′). To confirm the *hgfa* and *met* expression patterns, we searched a public zebrafish thyroid single-cell sequencing database (https://sumeet.shinyapps.io/zfthyroid/) and found that *hgfa* was expressed in thyroid stromal cells[22]; however, *met* was expressed in only some thyroid cells[22], which was consistent with

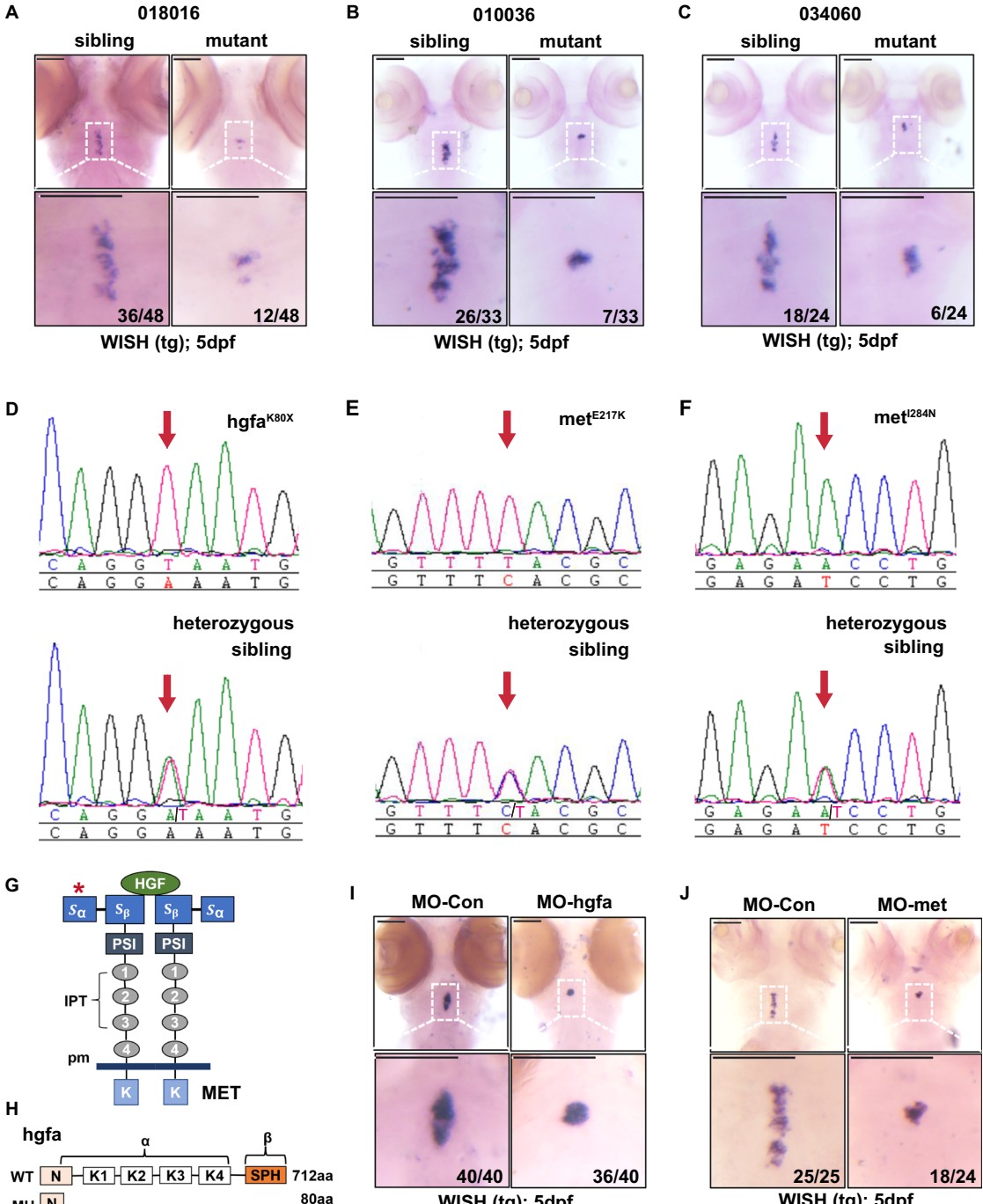

**Fig. 1 | Identification of hgfa and met mutations from three zebrafish lines with abnormal thyroid morphology by whole-exome sequencing and positional cloning. A–C** Three lines with abnormal thyroid morphology were screened by ENU. Thyroid marked by *tg* in 5 dpf larvae using WISH. Nearly 25% of the progeny from intercrossed heterozygotes exhibited an abnormal small thyroid morphology. Siblings contained wild-type and heterozygous larvae. Three independent experiments were carried for **A–C. D–F** Three mutants from the three zebrafish lines had mutations in *hgfa* and *met* that caused by K80X, E217K, and I284N amino acid substitutions, respectively. Verification by Sanger sequencing. **G** Diagram of the HGF/Met axis showing the mutations of Met localized in the Sema α domain. **H** Structures of the wild-type and mutant hgfa proteins. **I, J** Embryos injected with translation-blocking *hgfa* morpholinos (MO-*hgfa*) or *met* morpholinos (MO-*met*) to knock down *hgfa* or *met* induced the phenotype of abnormal small thyroid morphology at 5 dpf, as detected by in situ hybridization using *tg* as a probe. Three independent experiments were carried for **I, J.** MO-Con, embryos injected with standard control morpholino. Scale bars: 100 μm.

our results (Fig. 2B–D). We further analyzed the single-cell RNA-seq data of mouse thyroid generated in our lab[23], and found that fibroblasts were the main source of *Hgf* in thyroid tissue and that there was almost no *Hgf* expression in endothelial cells (cdh5, a specific marker of endothelial cells). These data suggested that the expression of *Hgf* was more abundant in fibroblasts than in endothelial cells in thyroid tissue (Supplementary Fig. 2). We thus presumed that the hgfa derived

from fibroblasts, rather than from endothelial cells of the surrounding vasculature in thyroid tissues, might play a key role in bifurcation during thyroid development.

## The p.I284N and p.E217K mutations impair Met activation

The extracellular portions of Met family members are composed of three domains, the N-terminal Sema domain, PSI domain and IPT

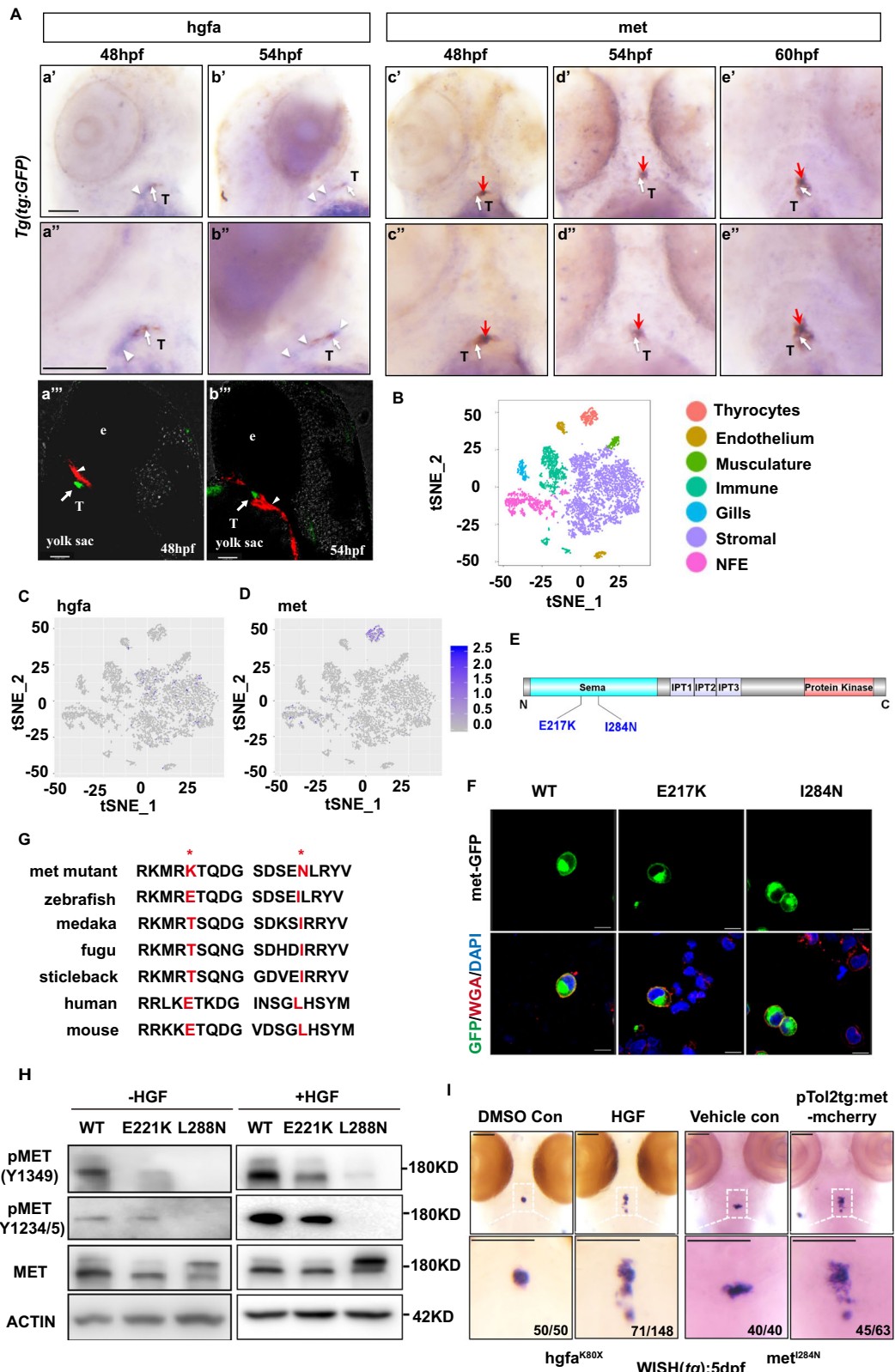

domain. The kinase domain in the intracellular portion of MET is connected to its extracellular domain by the transmembrane helix[24]. The p.I284 and p.E217 are localized in the Sema domain, within which MET undergoes proteolytic cleavage upon ligand binding (Fig. 2E). To determine whether the met[I284N] and met[E217K] mutations affect the distribution of met in the membrane, the p.I284N and p.E217K mutations in zebrafish *met* were overexpressed in HEK293T cells, and we found

that the p.I284N and p.E217K mutations did not affect the location of met in the membrane (Fig. 2F). Because no antibodies for met or phosphorylated met in zebrafish were available and the met[I284N] and met[E217K] substitutions in zebrafish mutants were relatively conserved in vertebrates (Fig. 2G), we generated human MET L288N and E221K mutants to investigate the effect of these mutations on the function of the met signaling pathway in the TOV112D cell line, which expressed

**Fig. 2 | The orthologous human MET mutations, I284N and E217K, impair HGF signaling. A** The spatiotemporal expression of *hgfa* and *met* in zebrafish embryos. *hgfa* is expressed around the thyroid primordium at 48 hpf and 54 hpf; *met* is colocalized in a portion of thyrocytes with *tg* in the embryos at 48–60 hpf, as detected by WISH. *hgfa* mRNA: blue, arrowhead on **panel A**; *met* mRNA: blue, red arrow on **panel A**; T: thyroid; tg:GFP caax: brown, white arrow on **panel A**. *hgfa* and *met* mRNA were detected by WISH, and thyrocytes were marked by immunohistochemical staining for GFP. a', b', a", b": left lateral position; a'", b'": right lateral position; c'–e' and c"–e": dorsal position. Scale bars: 100 μm. e:eye; *hgfa* mRNA was detected by fluorescence in situ hybridization, and thyrocytes were marked by immunofluorescence staining for GFP in Aa'" and Ab'". Three independent experiments were carried for **A**. **B–D** A single-cell atlas of the zebrafish thyroid gland showing that *hgfa* was expressed in some stromal cells but *met* was restrictedly expressed in thyrocytes (https://sumeet.shinyapps.io/zfthyroid). **E** Mutation sites in the secondary structure of the MET protein. Diagram of MET showing that the E217K and I284N substitutions are localized to the semaphorin-like domain (Sema), which contains the furin cleavage site (residues 302–307). **F** Immunofluorescence staining of HEK293T cells transfected with the zmet^WT, zmet^I217K and zmet^I284N plasmids, respectively. WGA staining (red) marks the cell membrane, and DAPI staining (blue) marks the nuclei. Three independent experiments were repeated with similar results. Scale bars: 100 μm. **G** The E217K and I284N substitutions in the mutants represent a significant shift in amino acids that are relatively conserved in vertebrates. **H** Western blotting was performed to analyze the expression and phosphorylation of MET in TOV112D cells transfected with plasmids containing human MET^WT, MET^E221K or MET^L288N after treatment with or without the human HGFα protein. Three independent experiments were repeated with similar results. **I** The rescue effects of human HGF factors or zebrafish wild-type *met* injection (pTol2tg:met-mCherry) on thyroid development are displayed. DMSO con, embryos injected with dimethyl sulfoxide (DMSO). Vehicle con, embryos injected with pTol2tg:mCherry. Scale bars: 100 μm. Three independent experiments were carried for **I**.

MET intracellular components but could not activate the MET signaling pathway[25,26]. The Met receptor is a disulfide-linked heterodimer that is derived via cleavage of a 190 kDa precursor at a site between amino acid residues 307 and 308 in the Sema domain to produce an extracellular 50 kDa α-subunit and a longer 140 kDa β-subunit[27]. Interestingly, the human MET^L288N mutation strongly disrupted the maturation of the MET precursor and therefore led to most of the nascent polypeptide of the MET^L288N mutant, with a size of 190 kDa, being formed (Fig. 2H); however, most of the nascent polypeptide of the human MET^E221K mutant was cleaved to obtain the 140 kDa mature form in TOV112D cells, which was similar to the results in TOV112D cells transfected with the wild-type MET plasmid (Fig. 2H). Furthermore, compared to that of the human wild-type MET receptor, phosphorylation of the met receptor at Y1234/5 and Y1349 was mildly decreased in the MET^E221K mutant and was significantly decreased in the MET^L288N mutant in TOV112D cells (Fig. 2H). Notably, injection of the pTol2tg:zmet-mCherry plasmid at the one-cell stage leading to the specific expression of wild-type zebrafish met in thyrocytes significantly rescued TD in larvae at 5 dpf in the mutants of the zebrafish met^I284N lines (Fig. 2I). Moreover, in the hgfa^K80X line, injection of the HGF protein in the 48 hpf yolk sac significantly rescued the abnormal thyroid morphology in larvae at 5 dpf (Fig. 2I). These data suggested that the abnormal thyroid phenotypes were caused by the loss-of-function mutations in *met* or *hgfa*.

## Hgfa/met mutants show thyroid dysgenesis and hypothyroidism

At 48 hpf, the thyroid primordium had a globular appearance and adopted a midline position just anterior to the aortic sac and the point of aortic arch artery 1 bifurcation, and there was no difference in the morphology of the thyroid between the mutant embryos and wild-type embryos from these three lines (Supplementary Fig. 3A). In addition, *tg*, *tpo* and *nis* were specifically expressed in mature thyroid follicles. We found that the shapes of the thyroid tissues in the homozygous larvae at 5 dpf were also abnormally small in the hgfa^K80X and met^I284N lines, as indicated by the *tpo* and *nis* probes (Fig. 3A, B). However, the overall larval size was unaffected in the mutants at 5 dpf (Supplementary Fig. 3B–E). Moreover, we found that the parathyroid gland (marked by *gcm2*) and ultimobranchial gland (marked by *calca*), two organs derived from the pharyngeal endoderm, were unaffected in 3 dpf and 5 dpf mutants (Supplementary Fig. 3F, G). These data suggested that the mutant pharyngeal phenotype in the hgfa^K80X and met^I284N zebrafish might be restricted to the thyroid. We marked mature thyroid follicles with T4 by immunofluorescence at 7 dpf (Fig. 3C) and 5 dpf (Fig. 3D) and found that the numbers and total volumes of mature follicles in homozygous larvae of the hgfa^K80X and met^I284N lines decreased when compared with those in the wild-type larvae (Fig. 3E-H). EdU staining revealed that thyroid cell proliferation was relatively unchanged in the mutants (Supplementary Fig. 4). Interestingly, the concentrations of T3 (tri-iodotyrosine) and T4 (thyroxine) in the tissues of 1.5-month-old adult zebrafish with homozygous mutations of hgfa^K80X (Figs. 3I, 3K) and met^I284N (Fig. 3J, L) were lower than those in their littermate controls, but the TSH concentrations were greater in the hgfa^K80X (Fig. 3M) and met^I284N (Fig. 3N) mutants. Moreover, the number of thyroid follicles was lower in the 1.5 mpf zebrafish mutants than in the wild-type siblings of the met^I284N line, as shown by H&E staining of sagittal sections (Fig. 3O–Q). Notably, we also found that the distribution of thyroid follicles in the mutants in the met^I284N line was also different from that in the WT siblings, in which the follicles were mainly concentrated in the anterior position (Fig. 3O). In addition, some characteristic large follicles were found in the thyroid of the mutant zebrafish (Supplementary Fig. 3H). These data indicated that loss of function of hgfa and/or met could result in TD and contribute to hypothyroidism in zebrafish.

## Hgfa/met deficiency disrupts the thyroid primordium caudal elongation

Thyroid budding and relocalization during the early stage of thyroid development seemed undisturbed in the hgfa^K80X and met^I284N homozygotes because the expression of *nkx2.4b* and *foxe1*, two key transcription factors involved in early thyroid development, was normal in the homozygous embryos at 48 hpf in the hgfa^K80X and met^I284N lines compared with their littermate controls (Supplementary Fig. 5A). Moreover, by using 3D imaging of double transgenic Tg(*flk1:EGFP; tg:mCherry*) embryos, we found that the thyroid primordium at 55 hpf embryos of the hgfa^K80X homozygote exhibited a globular appearance and was located at a midline position just anterior to the bifurcation point of pharyngeal arch artery 1, which was very similar to that in the embryos of the WT littermates (Supplementary Fig. 5B). However, by conducting live imaging of the embryos of the Tg(*tg:EGFP*) transgenic zebrafish line crossed with the two mutants hgfa^K80X and met^I284N, we found that the thyroid primordium in the homozygous embryos of the two morphants hgfa^K80X and met^I284N failed to caudally expand along the pharyngeal midline during the late stage of thyroid development in embryos from 56 hpf to 72 hpf, which was dramatically different from that in the embryos of the wild-type siblings (Supplementary Movies 1–3, Fig. 4A). These data indicated that the Hgfa-met pathway likely regulates caudal elongation through a pair of hypobranchial arteries in zebrafish during late thyroid development.

## Late thyroid expansion is mediated by hgf/met-MAPK pathway

The invasive cell behaviors, such as motility and proliferation, activated by HGF/Met were mediated by multiple parallel downstream signaling branches of the MET pathway. Three key downstream signaling branches of MET, the PI3K, ERK, and STAT3 pathways, have been reported[25]. Therefore, the 2–3 dpf embryos were treated with

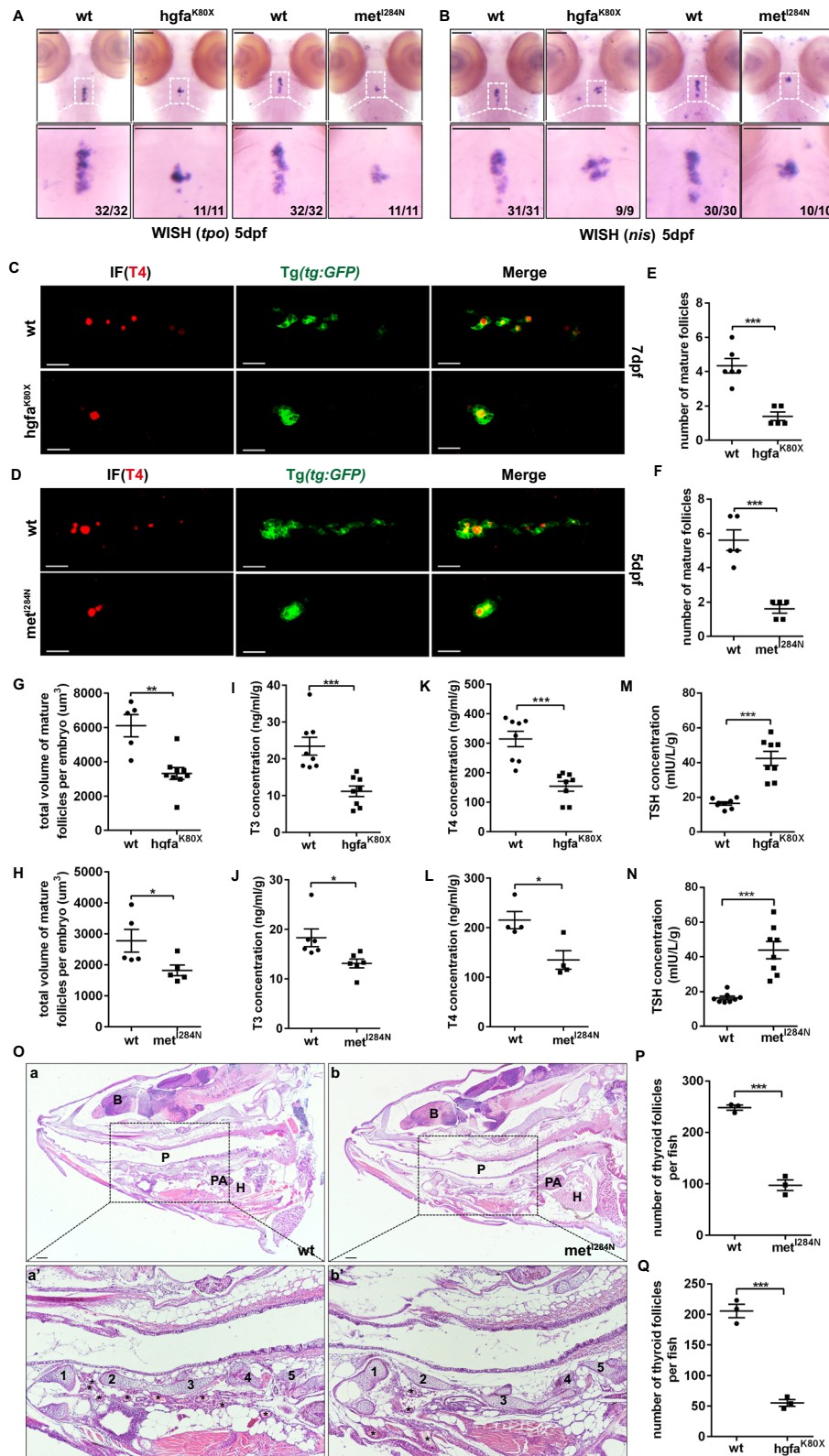

specific inhibitors of the PI3K, ERK and STAT3 signaling pathways, and we found that inhibition of ERK signaling with the MEK inhibitor U0126 induced a nearly perfect phenocopy of the small thyroid in 5 dpf larvae (Fig. 4Ba, 4Bb). However, treatment with the potent PI3K inhibitor LY294002 (Fig. 4Bc, 4Bd) or the STAT3/Src family kinase inhibitor SU6656 (Fig. 4Be, 4Bf) resulted in minor abnormalities in thyroid morphogenesis, while the p38 MAPK inhibitor SB203580 (Fig. 4B) had

almost no effect on thyroid development. We further treated the larvae with the ERK inhibitor U0126 at different stages of thyroid development, such as 6–10 hpf (gastrulation), 10–24 hpf (segmentation), 24–48 hpf (thyroid primordium development), 48–72 hpf (thyroid movement) and 72–120 hpf (Fig. 4C), and found that only if the embryos were treated with the inhibitor at 48–72 hpf was the thyroid morphology at 5 dpf similar to the phenotype of the hgfaK80X and

**Fig. 3 | The hgfaK80X and metI284N mutations result in hypothyroidism in zebrafish. A, B** The expression of *tpo* and *nis*, two specific thyroid markers, in wild-type or homozygous hgfa$^{K80X}$ and met$^{I284N}$ mutant zebrafish larvae at 5 dpf was detected via WISH. **C−H** Confocal examination of mature thyroid follicles marked by T4 immunofluorescence in the tg transgenic zebrafish line Tg*(tg:EGFP)* at 7 dpf **C** and 5 dpf **D. C, D**: ventral position, anterior is to the left. The total number **E, F** and volume **G, H** of thyroid follicles per larva were quantified. *n* = 6 for WT and *n* = 5 for hgfa$^{K80X}$ in E (*P* = 0.0003), *n* = 5 for WT and *n* = 5 for met$^{I284N}$ in F (*P* = 0.00027), *n* = 5 for WT and *n* = 9 for hgfa$^{K80X}$ in G (*P* = 0.0012), *n* = 5 for WT and *n* = 5 for met$^{I284N}$ in H (*P* = 0.0462). **I−N** The levels of thyroid hormones (T3, T4 and TSH) were detected via ELISA in 1.5-month-old WT and hgfa$^{K80X}$- or met$^{I284N}$- mutant zebrafish. The three individual zebrafish were pooled. After homogenization, the T3 and T4 concentrations in the supernatant were measured. Each group was analyzed in triplicates. *n* = 8 for each group in I (*P* = 0.00068), K (*P* = 0.00013), M (*P* = 0.00002), N (*P* = 0.00008); *n* = 6 for each group in J (*P* = 0.0268); *n* = 4 for each group in L (*P* = 0.01999). **O** Hematoxylin and eosin staining of sagittal sections of thyroid follicles from 1.5-month-old WT and met$^{I284N}$- mutant zebrafish. **P, Q** Quantification of the number of thyroid follicles in 1.5-month-old WT and mutant zebrafish. *N* = 3 for each group (*P* = 0.0002 for **P**, *P* = 0.00026 for **Q**). *B*, brain; *p*, pharyngeal; PA, pericardial aorta; *H*, heart; 1–5, cartilage; *thyroid follicles. Scale bars: 100 μm. *represents *P* < 0.05, ***represents *P* < 0.001. Data are presented as the mean ± SEM. Group comparisons were performed with two-sided Student's *t* test. Source data are provided as a Source data file.

met$^{I284N}$ mutants (Fig. 4D). Consistent with these findings, live imaging revealed that the thyroid primordium in U0126-treated embryos failed to caudally expand along the pharyngeal midline during the late stage of thyroid development from 60 hpf to 72 hpf (Supplementary Movie 4). However, the thyroid morphology of 5 dpf larvae was almost normal in the other groups (Fig. 4D). Notably, restricted over-expression of constitutively active MEK in thyrocytes rescued the TD phenotype of the 5 dpf homozygous mutant larvae in both the hgfa$^{K80X}$ and met$^{I284N}$ lines (Fig. 4E, F). These results indicated that HGF/Met regulated the late thyroid expansion of the thyroid primordium via the activation of the MAPK pathway in 2–3 dpf embryos of zebrafish.

### HGF/Met affect *adherens* junctions of thyrocytes via ERK-snail

Previous studies reported that HGF induces cell spreading and disruption of cell–cell adhesion through the MET-MAPK-Snail-E-cadherin pathway to promote the scattering of MDCK cells[28]. Interestingly, the scattering and migration abilities of the thyroid cancer cell line TPC1 were greater after treatment with human HGF (Supplementary Fig. 6A−H). Moreover, the effect of HGF on the scattering and migration of TPC1 cells was abolished by knockdown of *MET* expression (Supplementary Fig. 6A, B, E, F) or treatment with the MEK inhibitor U0126 (Supplementary Fig. 6C, D, G, H). Snail, a well-known E-cadherin transcriptional repressor, is upregulated by HGF-Met signaling directly through the activation of the MAPK pathway in MDCK and HepG2 cells[28]. Consistent with these findings, the HGF-induced increase in *Snail* and decrease in E-*cadherin* were reversed in TPC1 cells after MET knockdown or treatment with U0126 (Supplementary Fig. 6I, J).

Interestingly, compared to wild-type embryos, E-*cadherin* was dramatically increased in thyrocytes of homozygous embryos in hgfa$^{K80X}$ mutants, and the effect of the *HGF* mutation on E-*cadherin* expression in the thyroid primordium could be rescued by injection of human HGF in the homozygous embryos (Fig. 5A). Similarly, the increase in E-*cadherin* in thyrocytes of met$^{I284N}$ mutants could be inhibited through injection of the zebrafish with the pTol2tg:met or pTol2tg:Snail plasmid (Fig. 5B). Moreover, we also found that E-*cadherin* expression in thyroid cells increased when larvae were treated with the MEK inhibitor U0126 (Fig. 5B). Notably, TD in 5 dpf larvae of hgfa$^{K80X}$ morphants was partly rescued by injection of the pTol2tg:Snail plasmid or treatment with the adhesion inhibitor cytochalasin B (Cyto B) (Fig. 5C, D).

### HGF/Met is needed for the bifurcation of thyroid primordium in mice

Furthermore, we investigated whether HGF/Met-dependent thyroid bifurcation is conserved in mice. We generated a thyroid-specific Met knockout mouse model (Supplementary Fig. 7). In mice, bifurcation of the midline thyroid primordium occurs at approximately embryonic st.11.5 (E.11.5), during which time the thyroid tissue extends bilaterally along the third pharyngeal arch arteries. TTF1 is expressed in the thyroid anlage in the early stages of thyroid embryonic development; therefore, we used TTF1 as a marker to analyze the extension length of the thyroid in transverse sections of E11.5 embryos (Fig. 6A). We used the bifurcation index to evaluate whether the thyroid tissue extended bilaterally along the third pharyngeal arch arteries. The index was defined as the ratio of the maximum transverse diameter to the vertical diameter of the thyroid primordium. As expected, the bifurcation index of the thyroid primordium was dramatically lower at E11.5 in *Met* conditional knockout (Met-CKO) mice than in their WT littermates (Fig. 6A-C). We subsequently found that the late bifurcation and bilobation process of the thyroid in the mutants had caught up with that in wild-type mice at E12.5 and E13.5 (Fig. 6D). Moreover, the thyroid of Met-CKO mice exhibited normal morphogenesis of the left and right lobes at E15.5 (Fig. 6D). During bifurcation, all migrating thyroid bud progenitor cells adhere together as a single body, which is defined as collective cell migration. Recently, a study reported that HGF-Met-dependent ERK activation in the leader cells of collective cell migration played key roles in promoting lamellipodial extension and specified the cells at the free edge as the leader cells[29]. Therefore, we detected ERK activation in thyrocytes at the free edge in WT and Met-CKO mice at E11.5. Interestingly, we found that, in WT mice, ERK activation in thyrocytes at the free edge (Fig. 6E) was significantly stronger than that in Met-CKO mice (Fig. 6E). Notably, the number of thyrocytes with pErk-positive nucleus at the leading edge of the thyroid primordium at E11.5 was significantly lower in the Met-CKO mice (circle in Fig. 6E) than that in the wild-type mice (circle in Fig. 6E). These findings indicated that ERK activation in the leader cells of collective cell migration was impeded in the thyroid primordium of Met-CKO mice. We further observed that the adult Met-CKO mice showed a normal size and shape of the thyroid gland (Fig. 6F, G); however, the thyroid function of the Met-CKO mice was compromised, as shown by elevated thyroid-stimulating hormone levels at postnatal day 30 (Fig. 6H, I).

## Discussion

In the present study, positional cloning was used to identify a stop-gain mutation at K80X of *hgfa* and two missense mutations at E217K and I284N of *met* from three zebrafish lines with TD. These zebrafish lines with hypothyroidism were screened from Tübingen zebrafish using ENU mutagenesis screening at our zebrafish facility[18]. The TD phenotype in the hgfa$^{K80X}$ and met$^{I284N}$ mutants was rescued by injection of the human HGF protein and the zebrafish pTol2tg:zmet-mCherry plasmid, respectively, suggesting that the loss-of-function in *hgfa* and *met* were probably the pathogenic genes in the three zebrafish lines. Coincidentally, HGF is the ligand of the tyrosine kinase receptor Met and stimulates epithelial cell proliferation, motility, morphogenesis and angiogenesis in various organs via tyrosine phosphorylation of Met[19,30,31]. Knockout of *Hgf* or *Met* in mice led to embryonic lethality with placental, liver and limb muscle defects[32,33]. However, in the three mutant lines, the zebrafish survived to adulthood. The difference between mice and zebrafish may be attributed to the presence of other homologs in zebrafish that compensate for the genetic defects.

During zebrafish thyroid development, polarization of the first single thyroid follicular structure takes place at 55 hpf in embryos, and it subsequently migrates along a pair of hypobranchial arteries to form a row of posterior follicles[8]. Notably, in our homozygous mutant

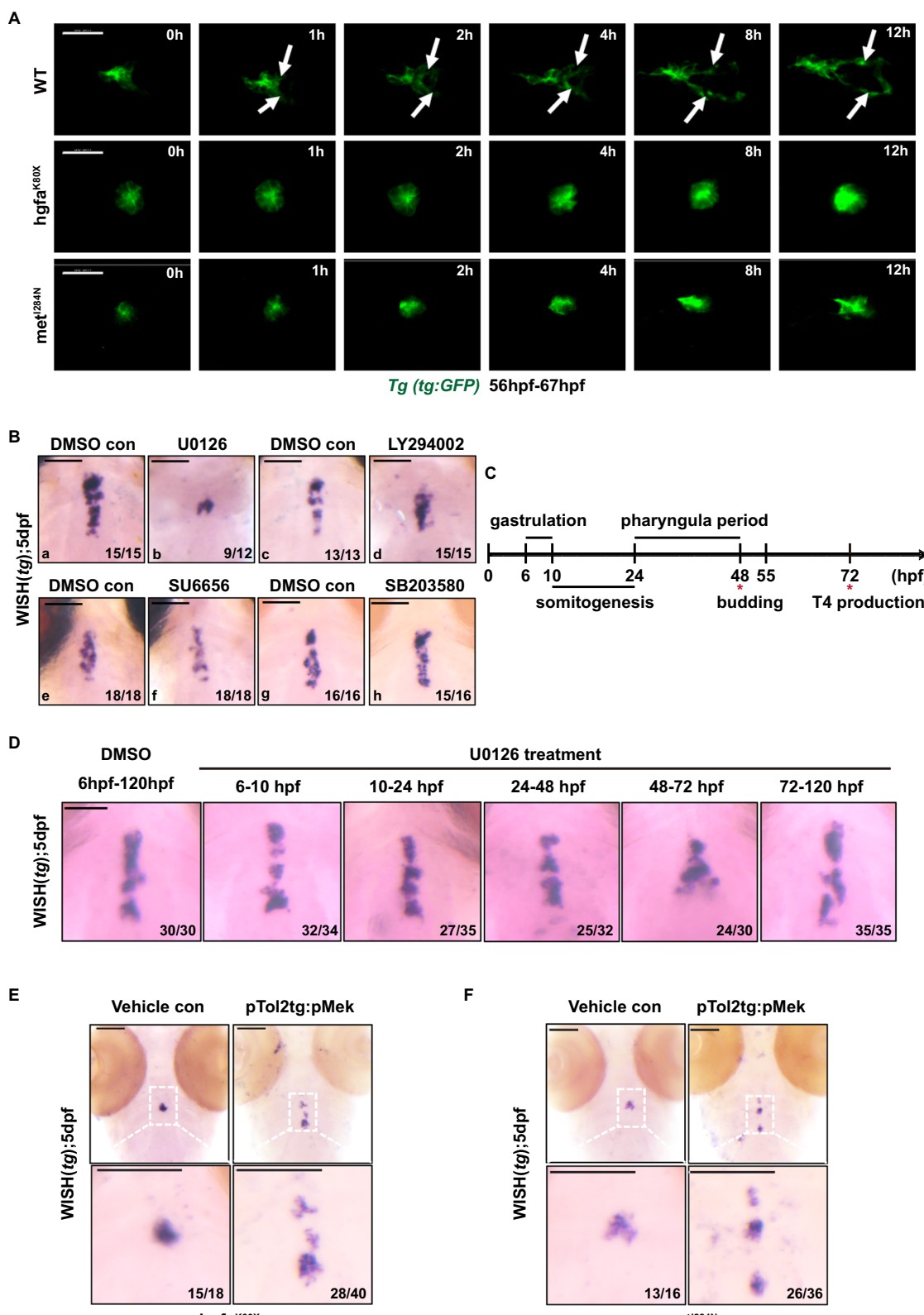

embryos of the hgfa[K80X] and met[I284N] lines, the size of the thyroid primordium in embryos before 55 hpf was seemingly not abnormal; however, in the late-stage mutant embryos, the thyroid progenitor cells gathered together and produced a condensed abnormal follicle without caudal elongation along the pharyngeal midline during late expansion of thyroid development (Supplementary Movies 1–3). Previous studies have shown that deficiency of the noncell autonomous Shh-Tbx1-FGF signaling pathway in mice leads to the failure of transverse elongation of the thyroid primordium along the third pharyngeal arch arteries after its descent[6,11–13], which is probably mediated by severe malformations of the vascular tree emerging from the outflow tract[14]. In this study, we found that the HGF/Met signaling pathway, as a cell-autonomous factor, plays an important role in regulating the normal bifurcation of the thyroid primordium in mice and zebrafish.

**Fig. 4 | Hgf/met promotes caudal expansion along the pharyngeal midline during the late stage of thyroid development via the MAPK signaling pathway. A** The Tg(tg:EGFP) transgenic zebrafish line was used to analyze the time lapse of thyroid development. Representative images chosen from Supplementary Movies 1–3 showing the differential thyroid phenotypes of wild-type and hgfa/met mutant embryos (ventral view, anterior is to the left). Scale bars: 30 μm. White arrows mark that thyroid primordium caudally expand along the pharyngeal midline in wild-type zebrafish. **B** Thyroid defects in small molecule-treated larvae. Inhibitors of MEK (U0126) successfully mimicked the small thyroid phenotypes in hgfa or met mutant embryos, while inhibitors of MAPK p38 (SB203580), PI3K (LY294002) and STAT3 (SU6656) had little effect on thyroid development. **C, D** WT embryos were treated with the MEK inhibitor U0126 during the different developmental stage of the zebrafish embryos. In **panel C**, the treatment periods of gastrulation and somitogenesis cover developmental periods preceding thyroid anlage formation (approximately 24 hpf), whereas the treatment period in the pharyngula represents the developmental period after the onset of thyroid anlage formation. The treatment period from 48–72 hpf covers the developmental period of during which the thyroid primordium forms branches. The treatment period during 3–5 dpf covers the developmental period of thyrocyte proliferation. In **panel D**, the thyroid primordium was detected by WISH using tg as a probe in 5 dpf larvae after treatment with U0126. Scale bars: 100 μm. **E, F** The rescue effects of continuously phosphorylated MEK (pTol2tg:pMek) on thyroid development in hgfa^K80X **E** and met^I284N **F** homozygous embryos were visualized via WISH. Scale bars: 100 μm. Three independent experiments were carried for **B, D–F**.

Deficiency of jag1b, a ligand of the Notch pathway, in zebrafish affects thyroid primordium elongation along the pharyngeal midline in both 72 and 120 hpf larvae, eventually leading to hypoplastic tissue with a progressive and significant reduction in tg-positive cells[34]. Interestingly, Notch and Shh-Tbx1-FGF signaling might also govern thyroid morphogenesis in the early stage of development[6,34]. However, hgfa-met signaling likely specifically regulates the late expansion of the thyroid primordium, and it is seemingly dispensable in the early phases of thyroid development. Notably, we found that the bifurcation of the thyroid primordium was significantly delayed in Met-CKO mice at the early or late E11.5 compared with that in wild-type mice. However, the late bifurcation and bilobation processes of the thyroid in Met-CKO mice had caught up with that in wild-type mice at E12.5 and E13.5 (Fig. 6D). Moreover, the thyroid of Met-CKO mice exhibited normal morphogenesis of the left and right lobes at E15.5 (Fig. 6D). These data suggested that the HGF/Met pathway likely specifically regulates the bifurcation of the thyroid primordium, but does not influence bilobation. In fact, thyroid development from E12.5 onwards in mice was regulated by the Fgf10 pathway, which corresponds to bilobation during thyroid devolpment[35].

In zebrafish embryos, a single thyroid follicular structure develops first, and later, it migrates along a pair of hypobranchial arteries to form a row of posterior follicles. Accordingly, in mice, the shape of the thyroid primordium changed at approximately E11.5, from a rounded ball to an elongated bar that projected bilaterally[36]; however, the mechanism of this process is still unclear. In this study, we found that the HGF/Met pathway is associated with this bifurcation process. The signal transduction of the HGF/Met pathway was mediated by three parallel downstream signaling cascades; ERK, PI3K and STAT3. In a previous study, PI3K and STAT3 were shown to be critical for Met-mediated exocrine morphogenesis during the formation of the pancreatic tail, while MAPK/ERK was not the primary mechanism driving pancreatic tail outgrowth in zebrafish[25]. However, in our study, the MAPK/ERK signaling branch played an important role in Met-mediated thyroid morphogenesis based on the results of treatment with specific inhibitors and rescue experiments with constitutively active MEK in zebrafish embryos. Recently, Hino et al. reported that during collective cell migration, ERK activation in leader cells was sustained in an HGF-dependent manner, but in follower cells, ERK activation was oscillatory waves in an epidermal growth factor signaling-dependent manner[29]. Moreover, HGF-dependent ERK activation in cells at the free edge of collective cell migration promoted lamellipodial extension[29]. Correspondingly, we found that ERK activation in the bilateral free edge thyrocytes was significantly greater in WT mice than in Met-CKO mice. These findings suggested that ERK activation in bilateral free edge thyrocytes promoted bifurcation of the thyroid primordium via HGF-Met signaling. E-cadherin, a calcium-dependent transmembrane protein involved in homotypic cell–cell interactions, allows neighboring cells to stick together. This adhesive molecule plays a central role in the development and maintenance of epithelial morphology, cell differentiation and migration[37]. Through time-lapse in vivo observation of thyroid morphogenesis in zebrafish, we found that thyrocytes in the thyroid primordium escape from the first anterior follicle after 55 hpf and elongate along the pharyngeal midline to form the remaining posterior follicles[20]. In the present study, we found that deficiency of the HGF/Met signaling pathway increased the expression of E-cadherin in thyrocytes and hindered the detachment of thyrocytes from the first anterior follicle to elongate along the pharyngeal midline, further leading to highly adhesive thyroid morphology and hypothyroidism in hgfa/met-deficient zebrafish embryos. Moreover, the aberrant late thyroid expansion and thyroid morphology induced by hgfa/met deficiency were partially rescued by downregulating the expression of E-cadherin in thyrocytes in 5-dpf zebrafish larvae via the injection of constitutively active Snail, a downstream signaling molecule of hgfa/met, which inhibited the expression of E-cadherin, as well as by treatment with the inhibitor Cyto B, disrupting the formation of adhesion complexes. These data suggest that the deficiency of HGF/Met leads to hypothyroidism and that TD might be mediated by a failure to downregulate the high expression of E-cadherin in thyrocytes through ERK activation, which further disturbs thyroid bifurcation.

## Methods

### Study approval

All studies were conducted according to the guidelines approved by the ethics committee of Shanghai Ninth People's Hospital (Approval No. SH9H-2022-A892-1).

### Zebrafish husbandry

We used wild-type zebrafish with the Tübingen background. Zebrafish maintenance and staging were performed using standard protocols. The methods were carried out in accordance with the approved guidelines. Zebrafish were housed in the zebrafish center of Shanghai Ninth People's Hospital. 5–20 adult mixed-sex fish per liter were grown in water with temperature, pH, and conductivity monitored daily and maintained at 26–29 °C, 7–8, and 200–3000 micro-Siemens, respectively. Live brine shrimp in addition to standard diet were used to feed adult fish. Transgenic zebrafish lines flk1:eGFP and tg:eGFP were obtained from China zebrafish resource center. Tg(tg:mCherry) was created by our lab. The following zebrafish lines were used in this study as previously described: Tg(flk1:EGFP)[38], Tg(tg:EGFP) and Tg(tg:mCherry)[20].

### Mouse model

Wild-type (WT) mice were obtained from Cyagen Biosciences, China. All mice were housed in a pathogen-free environment with the temperature maintained at $23 \pm 2$ °C and relative humidity at 50–65% under 12 h light/dark cycle. Mice were fed with normal chow (cat:19123123, Beijing KeaoXieli Feed Co., Beijing, China) with free access to food and water. Mice were euthanized with a combination of $CO_2$ and cervical dislocation to guarantee the death of the animals. Exon 7 of the mouse Met gene was selected as the conditional knockout region. A floxed Met targeting vector was constructed such that an upstream loxP site in the intron preceded exon 7 and the other loxP site in the intron immediately downstream of exon 7

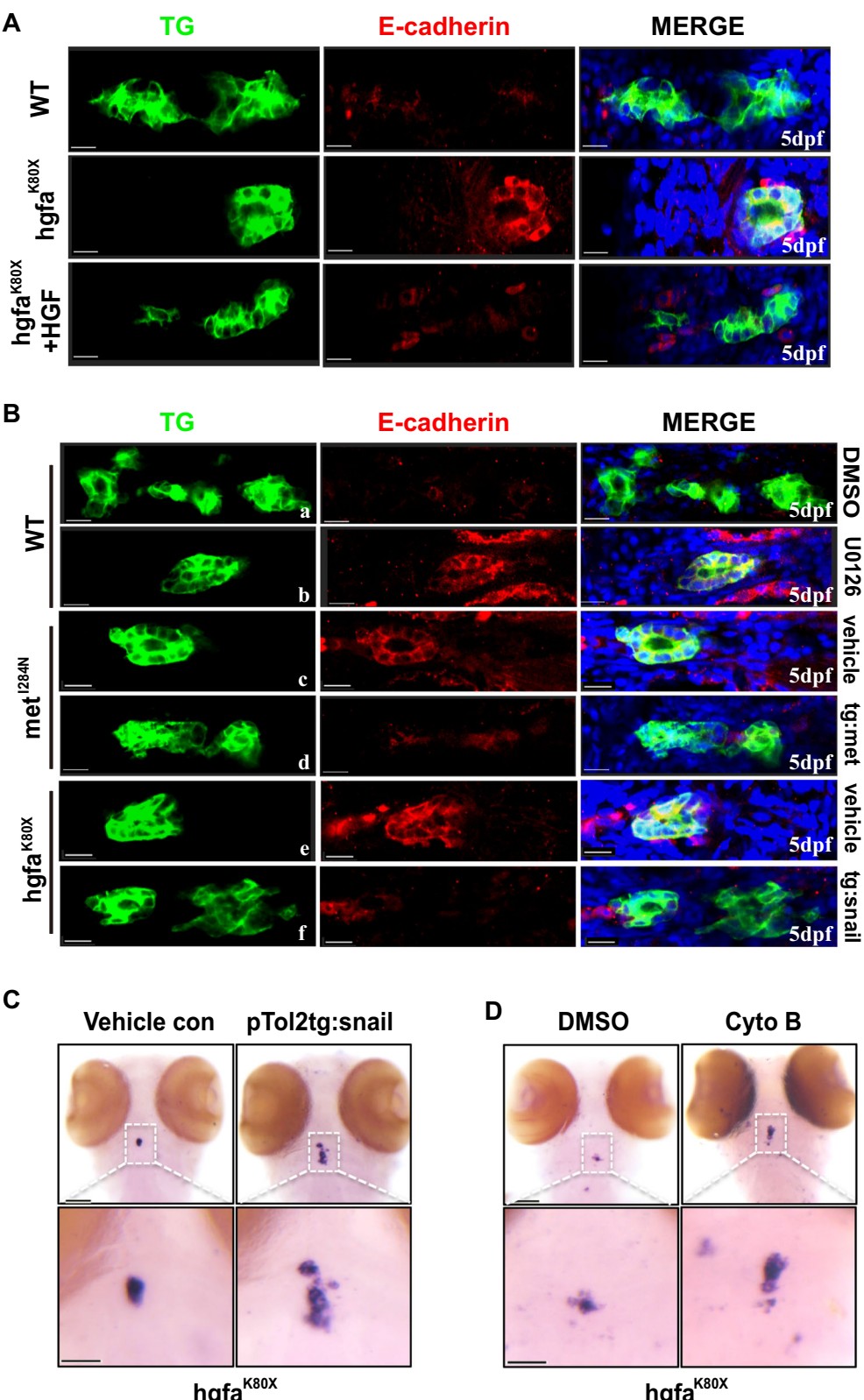

(Supplementary Fig. 7A). The targeting vector, gRNA and Cas9 were electroporated into fertilized eggs to generate chimeric animals. The *Met*[fl/fl] mice were crossed with Pax8-Cre transgenic mice[39] to generate thyroid-specific Met knockout mice (Supplementary Fig. 7A). The established mouse lines were maintained in a C57BL/6 background through brother/sister mating. Routine genotyping of the embryos was performed via PCR with tail DNA using the Met forward primer (F1)

5′- ATTTTAGCCTCGGAACAGTCTTCA-3′ and the reverse primer (R1) 5′-AATGAGAGATGTTTATCCTCAGTGG-3′. The expected sizes of the loxp allele and wildtype allele were 207 bp and 139 bp, respectively (Supplementary Fig. 7B). To examine Cre expression, the Cre (F) 5′-TACCT AGCCATGCCCTCACACCTAT-3′ and Cre (R) 5′-AGAGCCTGTTTTGC ACGTTCACC-3′ primers were used, generating a 537 bp fragment (Supplementary Fig. 7D). Thyroid-specific Met deletion was confirmed

**Fig. 5 | Hgf/met downregulates E-cadherin by activating MAPK-snail in vivo. A** Increased expression of E-cadherin in the thyroid primordium of *hgfa* homozygous mutant embryos was rescued by HGF injection. Larvae were in the ventral position, anterior to the right. **B** The expression of E-*cadherin* in the thyroid primordium of 5dpf zebrafish larvae under different conditions, including WT larvae treated with U0126 (b), *met* homozygous mutant embryos rescued by injection of pTol2tg:met (d), and *hgfa* homozygous mutant embryos rescued by injection of pTol2tg:Snail (f). Larvae were in the ventral position, anterior to the left. **C** The abnormal thyroid primordium in 5 dpf *hgfa* homozygous mutant larvae was rescued by specific overexpression of human Snail (pTol2tg:Snail) in thyrocytes. Vehicle con: 5dpf *hgfa* homozygous mutant larvae injected with pTol2tg:mCherry. **D** The rescue effects of the actin polymerization inhibitor cytochalasin B (cyto B) on the abnormal thyroid primordium in 5 dpf *hgfa* homozygous mutant larvae of zebrafish, detected by WISH in 5 dpf larvae. DMSO: 5dpf *hgfa* homozygous mutant larvae incubated with dimethyl sulfoxide (DMSO). Scale bars: 30 μm for **A, B** and 100 μm for **C, D**. Three independent experiments were repeated with similar results.

by the use of the forward primer F1 and reverse primer (R2) 5′-TTT GAAGTCTGAAGGGAAAGGTCC-3′, which generated a 298 bp fragment (Supplementary Fig. 7E). We isolated thyroid lobe tissue from adult mice and extracted total RNA. One microgram of total RNA was reverse transcribed to cDNA using a PrimeScript™ RT Reagent Kit with gDNA Eraser (TaKaRa). Amplification was carried out using SYBR Premix Ex Taq (TaKaRa) on the ABI QuantStudio 12 K Flex Real-Time PCR System (Life Technologies). Relative gene expression was normalized to that of GAPDH. The recombination efficiency was determined by the Met qPCR primer F (5′-GACCTTAAGCGAGAGCACGA-3′) and the qPCR reverse primer R (5′-ATGCACTGTATTGCGTCGTC-3′).

**Whole-mount in situ hybridization and immunohistochemistry**
Antisense digoxigenin-labeled RNA probes for *tshba, tg, gcm2, calca, nis, tpo, nkx2.4b, foxe1, met,* and *hgfa* were transcribed in vitro using SP6 or T7 polymerase (Ambion) with the DIG-RNA Labeling Kit (Invitrogen) from linearized constructs containing zebrafish *tshba, tg, gcm2, calca, nis, tpo, nkx2.4b, foxe1, met,* or *hgfa*. The primers used are shown in Supplementary Table 4. To detect *hgfa* and *met* expression in thyrocytes, after completion of in situ hybridization, the embryos were washed three times with PBST (PBS + 0.1% Tween 20) and reblocked for 2 h. Then, the samples were incubated with primary antibody (rabbit anti-GFP, Invitrogen, A-11120) overnight at 4 °C. After washing four times with PBST, the embryos were stained with secondary antibody (goat anti-rabbit HRP; Beyotime, A0239) and washed 3 times with PBST, after which the color was developed with DAB solution. Whole-mount specimens were observed and photographed under an SMZ25 dissecting microscope (Nikon) and documented using NIS-Elements BR 4.50.00 software (Nikon). To present the expression pattern of *hgfa* more clearly, we used the double fluorescence for WISH and immunofluorescence. For fluorescence of WISH, the process before blocking was the same as that for WISH, and then the embryos were incubated with an anti-digoxigenin-POD antibody at a 1:500 dilution in blocking solution overnight at 4 °C. After washing six times with PBST, the embryos were incubated for 1 h in TSA Plus Cy3 solution. After staining was terminated, the embryos were blocked in blocking solution (PBS + 0.3% Triton X-100 + 1% DMSO + 10 mg/mL bovine serum albumin +10% normal goat serum) for 2 h at RT. Then, rabbit anti-GFP antibodies were added to the blocking buffer, and the embryos were incubated at 4 °C overnight. The samples were washed three times with PBST, incubated with secondary antibodies at 4 °C overnight, and rinsed three times with PBST. Finally, images were captured by confocal microscopy.

**Genetic screening**
ENU mutagenesis screening was executed as standard protocols[40]. Briefly, 42 males (F0) were exposed to 3.5 mM ENU for 1 h at weekly intervals. Four weeks after ENU treatment, males were crossed with wild-type females at weekly intervals, and the progeny originating from mutagenized premeiotic germ cells obtained were raised (F1). All the F1 individuals were crossed with wild-type males or females to generate F2 lines. For screening, approximately 10 pairs of fish per F2 line were incrossed for egg laying, and we collected embryos and fixed larvae at 5 dpf. The zebrafish thyroid was marked through WISH with the *tg* probe.

**DNA preparation and Illumina whole-exome sequencing**
The pair of heterozygous fish that produced the small thyroid mutants were crossed again to generate homozygous mutants and their siblings. After WISH, we separated 20 pooled affected embryos and 20 pooled nonaffected sibling embryos. Genomic DNA from these two pools was prepared using an AxyPrep Multisource Genomic DNA Miniprep Kit (Axygen, USA) and quantified using Qubit 4.0. Whole-exome libraries were generated with 3 μg of genomic DNA and constructed using the SureSelect Target Enrichment System for Illumina Paired-end Multiplexed Sequencing Library Kit (Agilent Technologies, USA) according to the manufacturer's protocol.

**Genetic mapping**
Bulk segregant analysis (BSA) was performed using pooled DNA extracted from 20 siblings or small thyroid mutant embryos. First, Illumina reads were aligned to the reference zebrafish genome (zv9) by implementation of the Burrows–Wheeler Aligner (BWA)[41]. Only uniquely mapped single reads or confidently mapped paired-end reads were retained after alignment. Next, variants, including single-nucleotide variants (SNVs), were realigned with the Genome Analysis Toolkit (GATK) software version 4.1.4.0. Called variations from these two pools were filtered with the following criteria: (i) only nucleotides with a Phred quality of 50 or greater were considered; (ii) only reads with a mapping quality ratio equal to 0 (MQ0) and a depth less than 4 were considered; (iii) only nucleotides with an MQ greater than 40 were considered; (iv) only variants with a base quality greater than 30 were considered; (v) a minimum of 6× coverage and maximum of 5000× coverage in the mutation variants were needed; and (vi) only variants with a QD (qual by depth) > 2.0, FS (Fisher Strand) < 60.0, MQRankSum (Mapping Quality Rank Sum Test) > −12.5 and haplotype score < 13.0 were included. We combined variants of the sibling pool and mutant pool by combining GVCFs and only retained SNPs. We calculated the allelic frequency by the Euclidean distance (ED) and identified the region where the mutation was located as well as a list of putative coding region mutations in the linked genomic segment[21]. Moreover, we filtered variants from the mutant pools against the exome sequencing data of 15 zebrafish lines from our sequencing center. Then, we eliminated variants with a variant allele fraction (VAF) < 0.7 in the mutant pool or variants with a VAF > 0.5 in the sibling pool. We screened many SNPs through 15-line filtering, and combined with BSA analysis, the regions were further narrowed down to a few sites on a chromosome.

**PCR and genotyping**
Genomic DNA was extracted from whole larvae after WISH using lysis buffer in a thermal cycler. The conditions for this procedure were 10 min at 98 °C and a hold at 4 °C. Then, the larvae in lysis buffer were digested with proteinase *K* solution (100 μg/ml) overnight at 55 °C in a water bath. For F3 embryos from the met[I284N] and met[E217K] lines, each embryo was placed into one tube, and for F3 embryos from the hgfa[K80X] line, three were pooled into one tube. PCR was performed using Taq DNA polymerase (LaiFeng) with the following genotyping

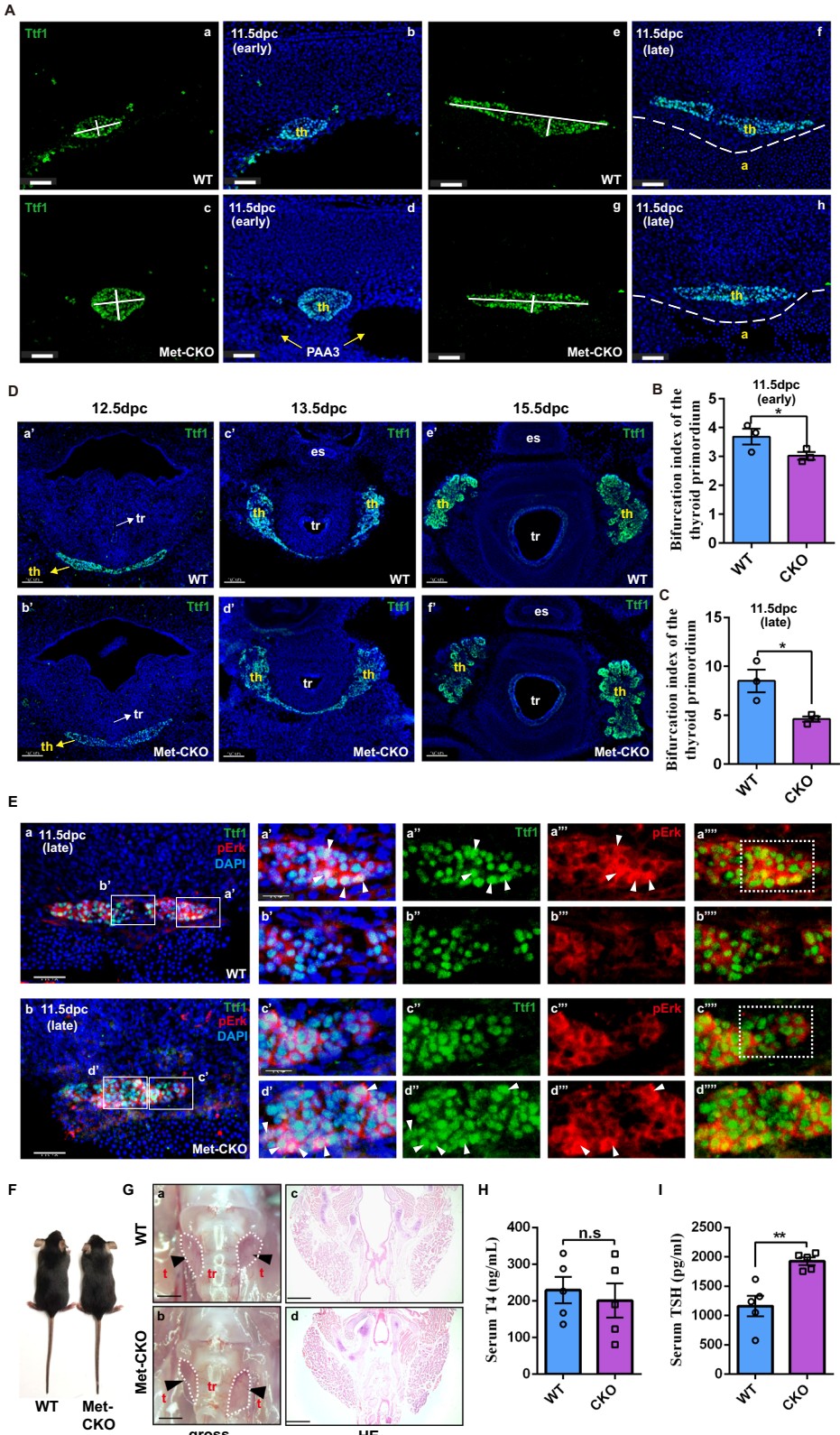

primer pairs: forward: met[I284N] genotype-F 5′-TTGAGTCAAACGGTCC ACGT-3′; met[I284N] genotype-R 5′- TTTCAAGCCCATCTCCTGCT-3′; met[E217K] genotype-F 5′- TAATGAGCGGTCGTGTTGTGA-3′; met[E217K] genotype-R 5′- AGCAGCCTGGAGAACGTTG-3′; hgfa[K80X] genotype-F 5′- CCTGGATGTCTGACCCAACG-3′; and hgfa[K80X] genotype-R 5′- CCAAA TGACTTCAGCATGTGC-3′. The PCR products were sequenced for genotyping.

### Single-cell RNA-seq data analysis

Raw fastq files were demultiplexed and mapped to the mouse mm10 reference genome using CellRanger (v3.0.1) recommended pipeline. The output of Cell Ranger is a raw count matrix for each sample, which records the unique molecular identifier (UMI) counts per gene and associated cell barcodes. Low-quality cells and genes were removed using Seurat V3.6.350 (http://satijalab.org/seurat/). We first filtered out

**Fig. 6 | HGF-MET-ERK signaling promotes the thyroid bifurcation in mice.**
**A** Compared to that in WT mice, bilateral elongation of the midline thyroid primordium in Met-CKO mice was delayed at E11.5. th: thyroid primordium; PAA3: third pharyngeal arch artery; a: aortic arch; scale bars: 50 μm. **B, C** Statistical assessment of the bifurcation index of the thyroid primordium in Met-CKO mice and WT mice at E11.5. The bifurcation index of the thyroid primordium was calculated as the ratio of the maximum transverse diameter to the maximum vertical diameter at the maximum section of the thyroid primordium in E11.5 embryos. $N = 3$ biologically independent samples ($P = 0.0477$ for **B** and $P = 0.0159$ for **C**). **D** Thyroid progenitors were identified by Ttf1 expression. There were no obvious abnormalities in the late bifurcation (E12.5), bilobation (E13.5) or lobe growth (E15.5) of the thyroid primordium in Met-CKO mice. th, thyroid primordium; es, esophagus; tr, trachea; scale bars: 50 μm. **E** Representative images showing the expression of pErk in thyrocytes in the maximum transverse section of the thyroid primordium in E11.5 embryos from WT and Met-CKO mice. The boxes in a′ and c′ indicate representative regions of lead cells at the free edge of the thyroid primordium in WT and Met-CKO mice, respectively; the boxes in b′ and d′ indicate representative regions of the following cells at the middle of the thyroid primordium in WT and Met-CKO mice, respectively. The arrowheads point to the thyrocytes with positive pErk. The dashed boxes in a‴ and c‴ represent the most different regions. Scale bars: 30 μm for a, b, 15 μm for (a′, c′). **F, G** Adult Met-CKO mice had a normal size and shape of the thyroid gland. t, thyroid lobe; tr, trachea. Scale bars: 800 μm for a, b, 200 μm for c, d. **H, I** Histograms showing the serum levels of T4 **H** and TSH **I** in one-month-old WT and Met-CKO mice. $N = 5$ biologically independent samples ($P = 0.321$ for H, $P = 0.0016$ for **I**). n.s represents not significant, *represents $P < 0.05$, **represents $P < 0.01$. Data are presented as the mean ± SEM. Group comparisons were performed with one-sided Student's $t$ test. Source data are provided as a Source data file.

low-quality cells that fit any of the following criteria: proportion of mitochondrial gene counts > 15%, UMIs < 500, or UMIs > 6000. Computational inference of doublets was performed by the DoubletFinder package in R. The expected doublet rate was set to 0.09. Doublets identified in each sample individually were excluded from the following analyses. After excluding low-quality cells and potential duplicates, we obtained 29962 cells in total: 5520 cells from 10 mice on postnatal day 5; 5520 cells from 10 mice on postnatal day 10; 6185 cells from 10 mice on postnatal day 20; and 12737 cells from 10 mice on postnatal day 30.

Data normalization, dimensional reduction, batch effect removal, clustering, and visualization were performed with the Seurat package. The four datasets were integrated using the SCTransform integration workflow on Seurat. The FindNeighbors() and FindClusters() function in Seurat were used for cell clustering. Meanwhile, shared nearest neighbor graph-based clustering was performed on the principal components analysis (PCA)-reduced data to identify the cell clusters. The resolution was set to 0.1 to obtain the major cell types of thyroid tissues. Seurat's DimPlot function was used to generate the UMAP (Uniform Manifold Approximation and Projection). The cell identity of each cluster was determined based on the expression of canonical marker genes. We identified endothelia (Flt1, Kdr, Ets1, Gata2), fibroblasts (Fn1, Dpt, Col1a1, Col1a2), myeloid cell(Itgam, Itgax), epithelia (Epcam, Tg), T cells (Cd3d, Cd3g, Cd4, Cd8a), B cells (Cd79a, Cd79b, Cd19), mastocyte (Kit, Il13, Gata3, and Rora), and neuron (Plp1), which represented all major cell types in thyroid tissues.

### Plasmid constructs
Zebrafish *met* (*zmet*) cDNA was transcribed from 2.5 dpf zebrafish embryo total RNA and subsequently inserted into pEGFP-N2 to construct the plasmid pEGFP-N2-zMET. The Tg promoter was cut from pTol2tg:mCherry (generously provided by Dr. Sabine, Université Libre de Bruxelles, Brussels, Belgium) by double digestion with BamHI and XhoI (Thermo Fisher, FD0054 and FD0694) and ligated into pTol2ubi: 2A-mCherry (generously provided by Dr. Yi Zhou, Boston Children's Hospital and Dana Farber Cancer Institute, Boston, USA) to construct pTol2tg: 2A-mCherry. The *zmet* gene was subsequently inserted into pTol2tg:2A-mCherry between the NotI and BamHI sites (Thermo Fisher, FD0053 and FD0594) to construct the plasmid pTol2tg:zmet-2A-mCherry. The *zmet* and *hMET* mutants (carried by the plasmid pLenti-MetGFP, Addgene, 37560) were constructed using PCR-based mutagenesis. Constitutively active human *MEK2* cDNA, cloned from the plasmid pMM9-MAPKK2-KW71[delta(48-55)/S222D/S226D] (Addgene, 40806), was inserted into pTol2tg:2A-mCherry to construct pTol2tg:pMEK2-2A-mCherry. Human Snail-6SA was cut from the GFP Snail-6SA plasmid (Addgene, 16228) by BamHI and NotI and inserted into pTol2tg:2A-mCherry to generate pTol2tg:Snail-6SA-2A-mCherry.

### Embryo microinjection of morpholinos, plasmid and HGF factor
Translation-blocking *met* and *hgfa* morpholinos were designed and purchased from Gene Tools LLC: *met* (ATG): 5′-CAGCTTCAGAAT AGTGAATTGTCAT-3′; *hgfa* (ATG): 5′-GTCCGATCTCATGTTTCTCGGT TTG-3′; and the standard control SCMO: 5′- CCTCTTACCTCAGTTAC AATTTATA-3′. The morpholinos were diluted in distilled water containing 0.05% phenol red and were injected at the 1-cell stage with 2 ng per embryo. Recombinant human HGF (MedChemExpress, USA) was injected into the yolk of 2 dpf embryos at 400 ng/mL, 2 nL per embryo. For transposon-containing vector overexpression in zebrafish, 100 ng/μL transposase mRNA and 30 ng/μl endotoxin-free plasmid were coinjected with 0.5 nL per embryo.

### Determination of the whole-body content of total thyroid hormones in zebrafish
The concentrations of total T4 (TT4), total TT3 and thyroid-stimulating hormone (TSH) were determined as previously described with a minor modifications[42,43]. Briefly, 1.5-month-old zebrafish were placed in Teflon tubes on ice and homogenized in 200 μl of ice-cold PBS by sonicating with 50% output for 20 s twice. The homogenized samples were centrifuged at 5000 g for 30 min at 4 °C, after which the supernatants were collected for thyroid hormone measurement. Three samples were pooled in a tube, and thyroid hormone concentrations were measured in duplicate using commercial enzyme-linked immunosorbent assay (ELISA) kits (Cloud Clone Corp., China; Product No. CEA453Ge for T3; and Product No. CEA452Ge for T4; CAMILO, Nanjing, China; Product No. 2F-KMLJf91039 for TSH) according to the manufacturer's instructions. For mice, TSH (Cloud Clone Corp., China; Product No. CEA463Mu) and FT4 hormone levels in blood collected from 1-month-old adults were detected.

### Zebrafish thyroid histology
Adult zebrafish were fixed in 4% PFA for 2 days, followed by decalcification in 20% EDTA for 1 week. The animals were cut at the mid-trunk level and processed in a Leica TP1050 tissue processor in preparation for paraffin embedding. The embedding station used was a Leica EG 1150H. Sagittal sections of the samples were cut at a thickness of 7 μm on a microtome (Thermo Fisher Scientific, microm HM325). Sections were arranged and fixed on Superfrost Plus™ slides (Thermo Fisher Scientific) and then subjected to hematoxylin and eosin (H&E) staining to evaluate the gross morphology of the thyroid and count thyroid follicles. For each group, three zebrafish were analyzed. All sections with follicles were used to determine the number of thyroid follicles.

### Immunofluorescence assay
For zebrafish, whole-mount immunofluorescence assays were performed as following. Briefly, embryos were fixed in 4% paraformaldehyde (PFA) overnight at 4 °C. Fixed larvae were washed in PBST and then dehydrated in graded PBST/methanol solutions (3:1, 1:1, 1:3) for 10 min each and stored in absolute methanol at -20 °C. Dehydrated embryos were rehydrated in graded PBST/methanol solutions for 10 min each, followed by washing three times with PBST for 5 min each

at room temperature (RT). Then, the embryos were treated with proteinase K and refixed with 4% PFA at 4 °C for 1 h. After washing three times, the embryos were blocked for 2 h at RT with blocking solution. Then, antibodies were added to the blocking buffer, and the embryos were incubated at 4 °C overnight. The samples were washed three times with PBST and incubated with secondary antibodies at 4 °C overnight, followed by rinsing three times with PBST. Finally, images were captured by confocal microscopy. For mice, staged embryos were obtained by dissection of pregnant females. The day on which the vaginal plug was detected was designated E0.5. The identification of genotypes was performed by tail DNA analysis. Embryos were fixed with 4% paraformaldehyde overnight at 4 °C before being cryoprotected in 30% sucrose, embedded in Tissue Tek OCT compound (Sakura Finetek, Alphen aan den Rijn, Netherlands) and frozen at −80 °C for storage. Serial horizontal sections (10 μm thick) were cut on a cryostat and collected on Superfrost Plus glass slides. The slices were washed three times in PBS and then blocked with 5% goat serum in PBS for 2 h at 37 °C. Then, the slices were incubated with primary antibodies diluted in blocking buffer overnight at 4 °C. After washing three times in PBS, the slices were incubated with secondary antibodies in blocking buffer. Finally, the sections were stained with DAPI and mounted with Fluorescence Mounting Medium. Immunofluorescence staining was carried out using the primary antibodies thyroxine (Abcam, ab30833), anti-CDH1 (BD), anti-GFP (Invitrogen), anti-TTF1 (Cell Signaling), and anti-pERK (Santa Cruz). The secondary antibodies used included goat anti-rabbit, Alexa Fluor 594 (Invitrogen), goat anti-mouse Alexa Fluor 594 (Invitrogen), goat anti-rabbit, and Alexa Fluor 488.

### Embryo treatment with chemicals

Wild-type embryos distributed in 12-well plates (20 embryos per well) were incubated with different chemicals from 48 to 96 hpf during the migration of the thyroid. The 5-dpf larvae were collected for WISH. The chemicals used were specific inhibitors of MEK (U0126, 10 μM), p38 MAPK (SB203580, 100 μM), PI3K (LY294002, 30 μM), and Src kinase (SU6656, 50 μM). Homozygous hgfa$^{K80X}$ and met$^{I284N}$ embryos were incubated with cytochalasin B (5 μM) from 48 to 96 hpf. Specific inhibitors were purchased from MedChemExpress. For EdU labeling, homozygous met$^{I284N}$ embryos and siblings distributed in 12-well plates (10 embryos per well) were incubated with EdU (20 μM, Thermo Fisher Scientific) from 48 to 96 hpf, and these 96-hpf larvae were collected for staining.

### Cell culture and reagents

TPC1 cell line was obtained from the Chinese Academy of Science in Shanghai. TOV112D cell line was obtained from FuHeng Cell Center. Human embryonic kidney (HEK293T) cell line was obtained from ATCC. Cells were cultured in a humidified incubator at 37 °C in the presence of 5% $CO_2$. TPC1 cells were cultured in RPMI-1640 medium supplemented with 10% fetal bovine serum (FBS; Gibco). HEK293T cells were maintained in DMEM supplemented with 10% FBS. TOV112D cells were cultured in a 1:1 mixture of MCDB 105 medium and medium 199 supplemented with 15% FBS. siRNAs targeting human *MET* (GenePharma) were transfected into cells using TransExcellent (CENJI). Wild-type and mutant pLenti-MetGFP were transfected into cells using Lipofectamine 3000 (Invitrogen).

### Cell immunofluorescence assay and western blot analysis

For cell immunofluorescence, HEK293T cells were plated on polypysine-coated coverslips and transfected with Lipofectamine with pEGPF-N2-zmet WT or site-directed mutagenesis-generated pEGPF-N2-zmet E217K and pEGPF-N2-zmet I284N. The cells were transfected for 48 h and then fixed in 4% paraformaldehyde, followed by incubation with wheat germ agglutinin conjugates 633 (WGA; Invitrogen) for cell membrane labeling. Then, the cells were incubated with rabbit anti-GFP antibody (Invitrogen) for 2 h prior to incubation with the goat anti-

rabbit secondary antibody, labeled with Alexa Fluor 488 (Invitrogen). For western blotting, protein samples were separated on 10% SDS-PAGE and transferred onto nitrocellulose membranes. The sections were blocked with 5% FBS (Sigma-Aldrich) in 0.1% TBST for 2 h at room temperature, incubated overnight at 4 °C with primary antibody, washed 3 times with TBST, and incubated for 2 h at room temperature with HRP-conjugated secondary antibody (1:5000). Following 3 washes with TBST, the membranes were developed with Immunoblot Forte Western HRP (Millipore), according to the manufacturer's instructions, and imaged using an AmershamTM Imager 600 (GE Healthcare). The antibodies used for western blotting were anti-Met (#8198), anti-phospho-Met (#3077 for Tyr1234/1235, # 3133 for Tyr 1349), anti-Erk (#9102), anti-phospho-Erk (#9101), anti-Snail (#3879; Cell Signaling), and anti-Cdh1 (BD).

### Cell scratch and scattering assays

Cell scratch assays were performed as previously described with minor changes[44]. In brief, TPC1 cells were plated in 6-well plates and cultured at 37 °C for 24 h. Cell scratches were made in the cell monolayer using a standard 200-μL pipette tip. The cells were washed three times with PBS to remove debris, after which they were incubated at 37 °C. Images were obtained 12 h and 24 h after wounding. The wound area was measured, and the percentage of wound healing was estimated using ImageJ software (NIH, Bethesda, MD, USA).

For cell scattering assays, TPC1 cells were allowed to grow into distinct colonies by seeding at a density of $2 \times 10^2$ cells/well in a 6-well plate. The majority of colonies consisted of approximately 40-60 cells at 48-72 h after seeding. The medium used to treat the colonies was then replaced with fresh medium supplemented with 10% FBS and 20 ng/ml HGF. If necessary, U0126 (10 μM) was added to the cells 2 h before HGF stimulation. The effect of HGF on cell scattering was evaluated at the indicated time points until 24 h, at which time the cells were photographed under a phase contrast microscope at 200× magnification. A colony was regarded as 'scattered' when half of the cells in the colony had lost contact with their neighboring cells and exhibited a fibroblast-like morphology. In total, 50 colonies were examined to quantify the extent of scattering[45].

### Statistical analysis

The data are presented as the mean ± standard error of the mean (SEM). Group comparisons of normally distributed data were performed with an unpaired Student's $t$ test. SPSS 19.0 software (IBM, Chicago, IL, USA) was used for all the statistical analyses. $P < 0.05$ indicated statistical significance. Diagrams were generated using GraphPad Prism 6.

### Reporting summary

Further information on research design is available in the Nature Portfolio Reporting Summary linked to this article.

## Data availability

All data associated with this study are presented in the paper and the Supplementary Materials or source data provided with this paper. The sc-RNA seq data used in this study are available in GEO database under accession code GSE231954. Source data are provided with this paper.

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

## Acknowledgements

This work was supported by National Science Foundation of China (grant number, 82300879 Y.F., 82070816 S.-X.Z., 82200874 F.S., 82200873 Q.-Y.Z.), Shanghai Science and Technology Committee (19140904200 M.D.), and Natural Science Foundation of Shanghai (22ZR1436600 B.H.).

## Author contributions

Conceptualization: H.S., M.D., and S.Z. Data curation: Y.F., J.W., S.S., Z.W., C.Z., L. Y., Q.Z., C.Y., F.W., S.L., F.S., and B.H. Formal analysis, Y.F., S.S and Z.W. Funding acquisition: Y.F., S.Z., F.S., Q.Z., D.M., and B.H. Visualization: Y.F., J.W. and S.S. Methodology: Y.F., J.W., S.S., Z.W., and M.D. Software: Z.W. Writing, original draft preparation: Y.F. Writing, review and editing: H.S., M.D. and Y.F. All authors have read and agreed to the published version of the manuscript.

## Competing interests

The authors declare no competing interests.
