## [Peer Review File · Nature Communications]

Deficiency of HGF/Met pathway leads to thyroid dysgenesis by impeding late thyroid expansionREVIEWER COMMENTS

Reviewer #1 (Remarks to the Author):

Fan et al. identify HGF/Met pathway to be important for thyroid formation. The authors identify the pathway in a forward genetic screen performed in zebrafish. The authors validate the role of the pathway in mouse, using a tissue-specific knockout. Overall, the experiments conducted by the authors support with the message provided in the manuscript. The identification of the pathway adds new player in thyroid gland morphogenesis, an understudied topic.

Comments:

1. The details related to quantification of thyroid follicular cells in adults (Fig. 3B) is missing. The figure legend says "Quantification of the number of thyroid follicles in 1.5-month-old WT and metl284N mutated zebrafish.". However, the figure shows "number of thyroid follicular cells per fish".
2. Similarly, in the same figure, for 5 dpf quantification (Fig. 3D, G), the Y-axis should be "number of follicles" rather than "number of mature follicular cells". This reviewer believes that the number of thyrocytes (thyroid follicular cells) at 5 dpf is ~60 (<https://www.nature.com/articles/s41598-018-24036-4>).
3. It would be good if the authors quantify the number of thyroid follicular cells at 5 dpf. And if possible the proliferation of thyrocytes. It could be possible that HGF/Met signaling regulates cell-cycle and thereby the shape and size of thyroid follicles.
4. Edits in summary:
Line 37: It's -> It has been
Line 40: "in vivo" italicise
Line 44: restored -> rescued
5. It would be good to cite the article related to the single-cell atlas of zebrafish thyroid used in Fig. 2 B - D. PMID: 33140917 PMCID: PMC7726803 DOI: 10.15252/embr.202050612
6. The nomenclature for zebrafish transgenic lines need to be checked: <https://zfin.atlassian.net/wiki/spaces/general/pages/1818394635/ZFIN+Zebrafish+Nomenclature+Conventions>
For instance, the line (Tol2:tg-met-mCherry), should be written as Tg(Tol2-tg:met-mCherry). And (Tg:(tg-GFP)) would be Tg(tg:GFP).
All transgenic lines are named as Tg(promoter:construct).

Reviewer #2 (Remarks to the Author):

By whole-genome genetic screening of congenital hypothyroidism in a large zebrafish population, authors identified three "families" of spontaneously developed inactivation mutations in the Hgf and Met genes, which by various functional analyses were linked to growth of the thyroid primordium. Results inferring a role of Hgf/Met pathway in thyroid developmental growth were also confirmed for mouse embryos.

Although of biological interest, and notably with use of advanced and sound methodologies of the fish part, there are several issues of concern in this manuscript that require consideration.

1. The bearing concept of comparing thyroid morphogenesis in zebrafish and mouse is flawed by the fact that in fish the thyroid does not form a solid organ but rather consists of a row of loosely connected follicles i.e. what is being studied is folliculogenesis of cells that are already functionally differentiated. Although this process takes place close to and is depending on pharyngeal vasculature it does not correspond to the the bilateral elongation manufactured by mainly undifferentiated progenitor cells during thyroid organogenesis in mice (and probably in all mammals including man that display a bilobed thyroid).

2. Blocked propagation of thyroid follicle formation in Hgf/Met deficient zebrafish related to EMT inhibition is interesting. However, no detailed information is provided on whether this effect depend on inhibited dissociation of cells from the first formed follicle or inhibited reassociation of cells that before undergo partial EMT. Moreover, in view of recent findings in mouse embryonic thyroid that folliculogenesis designating epithelial differentiation takes place much earlier than previously understood (Johansson et al, Front Endocrinol, 12:760541, 2021), comparison to mice should rather focus on folliculogenesis than proliferation/expansion of the thyroid primordium. In other words, it would be very interesting to learn if HGF/Met-dependent partial EMT might regulate thyroid folliculogenesis across species.

3. There is limited information on the extrathyroidal phenotype of mutants, and the potential influence of organogenesis as a whole. Although authors convincingly show that Hgf rescues follicle generation in mutants, the likely source of Hgf in vivo is the surrounding mesenchyme, supported by data provided in Fig. 2. Are overall embryo size and the development of major organ systems or, in particular, other anterior/pharyngeal endoderm derivatives (e.g. parathyroid and ultimobranchial gland) really unaffected in mutants? If mutant pharyngeal phenotype indeed is retracted to thyroid this should be documented.

4. Morphometric calculations of mouse thyroid shape and size (Fig. 6) infer a role of Hgf/Met pathway prior to E11.5, which represents the end of descensus and start of bilateral elongation of the midline thyroid primordium. What happens thereafter in mutants? Is thyroid phenotype further aggravated with impact on subsequent growth and glandular development, or do you see catch-up of cell proliferation and normal progression the bilobation process? Notably, Fgf10 regulates thyroid developmental growth from E12.5 and onwards (Liang et al Development 145:dev146829, 2018). Thus, it would should be possible to answer whether Hgf/Met action is limited to a previous developmental stage, as suggested by current zebrafish data, and also whether the mutant phenotype merely is the result of delayed thyroid development.

Normal S-T4 and only doubled S-TSH levels postnatally are consistent with subclinical hypothyroidism and argue that thyroid tissue volume and function essentially are sufficient, but is the gland smaller than in wildtype?

Minor:

- maybe mutant "families" should be rephrased to mutant strains or lines

- Figs. 1g and h might be excluded or at least reduced in size to conform with other.

- Analysis of TSH (thyrotropin) amounts should preferably be added as supplementary to Figs. 3i-l to estimate level (differences) of hypothyroidism among mutants.

- Legend to Fig. 3M is lacking key to abbreviations (PA and H) and numbers (1-5). Please check all legends for similar.

- Morphometric calculation of cell numbers (e.g. in Figs. 3d-h and n) should be indicated. Both Methods section and figure legend(s) are lacking this information.

- Figs. 5a-j based on cell line analysis does not add much and may be replaced by other more of interest to the paper's major content. The results are largely confirming previous wellknown effects of Hgf/Scatter factor and be removed in toto or switched to Supplementary.

- Fig. 6A, the IF images are generally of poor quality. Moreover, it is expected that phospho-ERK designating activation of the MAPK pathway should be translocated to the nucleus and captured by IF at least in some cells, which is not the case. This raises doubt on antibody specificity, alternatively on biological effect.

English in need of editing preferably by native.

Reviewer #3 (Remarks to the Author):

Summary of the Manuscript

The MS of Fang et al. describes defective thyroid gland development in three zebrafish mutant lines as a result of disruption of HGF-Met signaling. The mutant lines were previously generated by the same group in an ENU mutagenesis screen and were selected for further studies because of a unique thyroid phenotype. The group mapped the mutations to two loci encoding for *hgfa* and *met*, two genes that act in the same signaling pathway (hepatocyte growth factor signaling) as *hgfa* encodes a ligand for the receptor tyrosine kinase *met*. For both *hgfa* and *met*, the Zebrafish Information Network (ZFIN) database lists quite a number of mutant alleles but a thyroid developmental phenotype has so far not been reported. The authors provide limited data on the expression of the two genes showing expression within (*met*) or adjacent to (*hgfa*) the developing zebrafish thyroid primordium. A series of additional in vitro experiments confirmed that the *met* mutations p.I284 and p.E217 impaired *met* activity. Rescue experiments based on injection of WT *met* or WT *hgfa* could rescue the thyroid phenotypes in 5-day-old zebrafish.

The authors next characterized the thyroid phenotypes in some more detail showing that late phases of thyroid development are specifically affected, that defective thyroid morphogenesis results in hypothyroidism and that adult mutant fish have a reduced number of thyroid follicles. Live imaging of transgenic thyroid-specific reporter lines convincingly visualizes the defective expansion of the zebrafish thyroid.

The authors next used a small molecule screen and showed that inhibition of ERK but not PI3K or STAT3 intracellular signaling phenocopies the thyroid defects seen in mutant fish. In a very interesting section of the MS, the authors show that reduced cell adhesion is critical for thyroid expansion in zebrafish and that increased cell adhesion in mutants with perturbed HGF-Met signaling is likely one critical factor causing the thyroid defects.

In the final part of the MS, the authors describe limited phenotyping studies of conditional Met-KO mouse (thyroid-specific invalidation of Met). The data presented appear to show

impaired ERK signaling in the thyroid primordium of KO embryos but are not very convincing in the way the data were presented.

General comments

The identification of HGF-Met signaling as a critical pathway to regulate late expansion of the thyroid is a novel finding. To my knowledge, the HGF-Met pathway has not yet been linked to defects in thyroid development and function in mouse models or in the context of congenital hypothyroidism in human. A particular strength of the study by Fang et al. is the demonstration that HGF-Met signaling is linked to Snail-mediated regulation of cell-cell adhesion and that tight regulation of adhesion dynamics could be critical during certain phases of thyroid primordium expansion. I appreciate the attempts by the authors to translate their findings in zebrafish to mouse thyroid organogenesis to demonstrate the broader role of this morphogenetic process but the observations in embryonic mice are poorly presented.

Despite the novelty of the findings and the potential relevance of the reported data to better understand thyroid morphogenesis, the manuscript in its current form has not sufficient quality to recommend it for publication in Nature Communications.

Below, I summarize some of my major concerns.

Major concerns

The use of English language needs to be greatly improved. In several instances, a poor English prevented me from understanding what exactly the authors intended to describe. Unfortunately, in its current form, the MS is written for a very small community of readers interested or experienced in thyroid morphogenesis and the MS might fail to attract a broader readership. Due to my personal expertise in pharyngeal organogenesis in zebrafish and mouse embryos, I could follow most descriptions and explanations provided by the authors. However, when reading the MS, I had the strong expression that readers that are not familiar with zebrafish or mouse thyroid organogenesis might experience great difficulties in understanding the anatomical details or developmental processes described.

The MS would benefit greatly from short introductory notes how thyroid development proceeds (particularly in zebrafish) with some more details on anatomical landmarks that characterize the environment in which the thyroid primordium develops.

The authors should also improve the use of an appropriate terminology to describe anatomical details and morphogenetic processes. For example, avoid wording like: “dot-like” (line 91), “resembling a ball positioned at the bottom of the aortic arch,” (line 118), “scattering and migration abilities” and “from a rounded ball to an elongated bar that projected bilaterally”(line 373)

I also suggest not to use the term “congenital hypothyroidism” (line 211) in relation to zebrafish embryo models. Congenital hypothyroidism describes a condition of reduced circulating TH at birth, so the term is not applicable to zebrafish embryos.

I strongly disagree with the description of the main phenotype as a defect in “rostral” elongation. As the authors demonstrate neatly in their live imaging experiments, there is clearly a defect in the “caudal” expansion of the primordium. The apparent expansion/elongation of the normal zebrafish thyroid primordium between day 2 and day 5 is due to the addition of cells to the caudal portion of the primordium (as the movie nicely show). The many WISH images in Figure 1 also show that the caudal portion of the thyroid is missing.

Selected specific comments

I am not satisfied with the quality of some WISH experiments. Particularly the stainings of

hgfa and also of foxe1 are very weak and do not show all structures that the express these genes (i.e. there is labeling of the pharyngeal epithelium by the foxe1 probe). This comment relates to the statements by the authors that hgfa expression adjacent to the thyroid was only detectable between 48 and 55 hpf but the defects in thyroid expansion happened during the period from 55 and 120 hpf. This would imply that HGF-Met signaling is active only during the onset of thyroid expansion with lasting consequences even in the absence of continuous hgfa expression near the met-expressing thyroid.

Graphs in Figure 3D and 3G visualize the number of “mature follicular cells”. I see many more cells in the corresponding images in 3C and 3F. The same is true for the graph in Figure 3N. Please check.

Line 246 – 248: the authors note “These data indicated that the Hgfa-met pathway probably regulated rostral elongation along a pair of hypobranchial arteries in zebrafish during late thyroid development”. It would be very informative to visualize the relative positioning of the thyroid and the paired hypobranchial arteries in WT and mutant fish during the process of elongation. A live imaging video and some images in Figure S3B are provided but only for the period before elongation. With respect to the specific anatomical relationship between the pharyngeal vessels and the thyroid the question arises if it is the vessels that express hgfa. From a larger conceptual perspective, this is an extremely relevant question because if vessels are the source of hgfa, then HGF-Met signaling could also be compromised in cases of vascular maldevelopment.

Figure 6. I could not orient myself on the images shown in Figure 6. Are these really sagittal sections? How can a “bifurcation index” be calculated on sagittal sections? Please provide additional images that contain anatomical landmarks (such as the nearby vessels).

Reviewer Comments:

Reviewer #1 (Remarks to the Author):

Fan et al. identify HGF/Met pathway to be important for thyroid formation. The authors identify the pathway in a forward genetic screen performed in zebrafish. The authors validate the role of the pathway in mouse, using a tissue-specific knockout. Overall, the experiments conducted by the authors support with the message provided in the manuscript. The identification of the pathway adds new player in thyroid gland morphogenesis, an understudied topic.

Reply: We greatly appreciate your encouraging comments and for your suggestions. Thank you for your useful comments and suggestions. We have modified the manuscript accordingly, and detailed corrections are listed below point by point. Revised portion are marked in blue in the paper.

1) The details related to quantification of thyroid follicular cells in adults (Fig. 3B) is missing. The figure legend says "Quantification of the number of thyroid follicles in 1.5-month-old WT and metI284N mutated zebrafish." However, the figure shows "number of thyroid follicular cells per fish".

Response: Thank you for your carefully reviewing our paper. We are sorry for the careless mistake. We have revised the Y-axis of figure 3E,F,G,H,P,Q as number of thyroid follicles.

2) Similarly, in the same figure, for 5 dpf quantification (Fig. 3D, G), the Y-axis should be "number of follicles" rather than "number of mature follicular cells"). This reviewer believes that the number of thyrocytes (thyroid follicular cells) at 5 dpf is ~60 (<https://www.nature.com/articles/s41598-018-24036-4>).

Response: Thank you for your carefully reviewing our paper. We are sorry for the careless mistake. We have revised the Y-axis of figure 3E,F,G,H,P,Q as number of thyroid follicles.

3) It would be good if the authors quantify the number of thyroid follicular cells at 5 dpf. And if possible the proliferation of thyrocytes. It could be possible that HGF/Met signaling regulates cell-cycle and thereby the shape and size of thyroid follicles.

Response: Thank you for your valuable suggestions. As we have no transgenic fish line labelling thyrocytes nuclei, it is not easy to accurately quantify the number of thyrocytes. So we detected the proliferation of thyrocytes in wildtype and met mutant zebrafish embryos at 4 dpf by Edu staining and the results showed that thyrocytes proliferation in mutants remained relatively unchanged (Fig. S3).

4).Edits in summary:

Line 37: It's -> It has been

Response: Thank you for your carefully reviewing our paper. We have revised the mistake in the manuscript (line 55).

Line 40: "in vivo" italicize

Response: We felt sorry for our careless mistakes. We have revised the mistake in the manuscript (line 57).

Line 44: restored -> rescued

Response: Thank you for your carefully review. We have revised it in the manuscript (line 63).

5. It would be good to cite the article related to the single-cell atlas of zebrafish thyriod used in Fig. 2 B - D. PMID: 33140917 PMCID: PMC7726803 DOI: 10.15252/embr.202050612

Response: Thank you for your suggestion, we have cited the article related to the single-cell atlas of zebrafish thyriod used in Fig. 2 B (line 208-209).

6. The nomenclature for zebrafish transgenic lines need to be checked: <https://zfin.atlassian.net/wiki/spaces/general/pages/1818394635/ZFIN+Zebrafish+Nomenclature+Conventions>

For instance, the line (Tol2:tg-met-mCherry), should be written as Tg(Tol2-tg:met-mCherry). And (Tg:(tg-GFP)) would be Tg(tg:GFP).

All transgenic lines are named as Tg(promoter:construct).

Response: Thank you for your suggestions. We have revised the nomenclature for zebrafish transgenic lines and marked blue in the manuscript.

Reviewer #2 (Remarks to the Author):

By whole-genome genetic screening of congenital hypothyroidism in a large zebrafish population, authors identified three "families" of spontaneously developed inactivation mutations in the Hgf and Met genes, which by various functional analyses were linked to growth of the thyroid primordium. Results inferring a role of Hgf/Met pathway in thyroid developmental growth were also confirmed for mouse embryos.

Although of biological interest, and notably with use of advanced and sound methodologies of the fish part, there are several issues of concern in this manuscript that require consideration.

1) The bearing concept of comparing thyroid morphogenesis in zebrafish and mouse is flawed by the fact that in fish the thyroid does not form a solid organ but rather consists of a row of loosely connected follicles i.e. what is being studied is folliculogenesis of cells that are already functionally differentiated. Although this process takes place close to and is depending on pharyngeal vasculature it does not correspond to the \ bilateral elongation manufactured by mainly undifferentiated progenitor cells during thyroid organogenesis in mice (and probably in all mammals including man that display a bilobed thyroid).

Response:

Thank you for your valuable comments. The concerns of the reviewer #2 about that “the thyroid morphology in zebrafish is different from that in mice, and the differentiation of the thyrocytes is also different when folliculogenesis between zebrafish and mice, the folliculogenesis of thyrocytes that are already functionally differentiated in zebrafish” is correct. In fact, the thyroid follicles from zebrafish embryos at 55hpf to 72hpf could synthesized the thyroid hormone when we detected by whole-mount immunofluorescence assays using T4 antibody. We add the difference about the process of thyroid development in the introduction. (line 86-97).

2) Blocked propagation of thyroid follicle formation in Hgf/Met deficient zebrafish related to EMT inhibition is interesting. However, no detailed information is

provided on whether this effect depend on inhibited dissociation of cells from the first formed follicle or inhibited reassociation of cells that before undergo partial EMT. Moreover, in view of recent findings in mouse embryonic thyroid that folliculogenesis designating epithelial differentiation takes place much earlier than previously understood (Johansson et al, Front Endocrinol, 12:760541, 2021), comparison to mice should rather focus on folliculogenesis than proliferation/expansion of the thyroid primordium. In other words, it would be very interesting to learn if HGF/Met-dependent partial EMT might regulate thyroid folliculogenesis across species.

Response: Thank you for your valuable suggestions. In the revised manuscript, we furtherly observed the phenotype in Met-CKO mice at E11.5 to E15.5. The results showed that the bifurcation of thyroid primordium were significantly delayed in Met-CKO mice at the early or late E11.5 when compared with that in wildtype mice. However, the bilobation process of thyroid in Met-CKO mice had caught up with that in wildtype mice at E12.5 and E13.5 (line361-363 and Fig. 6Da'-6Dd'). The thyroid of Met-CKO mice had normal formation of the left and right lobes at E15.5 (line363-364 and Fig. 6De'-6Df'). These findings suggested the HGF/Met pathway play key roles in the bifurcation of thyroid primordium, but did not influence the bilobation. Of course, the questions about whether HGF/Met pathway regulating folliculogenesis is a very interesting and we will investigate this topic in the future.

3) There is limited information on the extrathyroidal phenotype of mutants, and the potential influence of organogenesis as a whole. Although authors convincingly show that Hgf rescues follicle generation in mutants, the likely source of Hgf in vivo is the surrounding mesenchyme, supported by data provided in Fig. 2. Are overall embryo size and the development of major organ systems or, in particular, other anterior/pharyngeal endoderm derivatives (e.g. parathyroid and ultimobranchial gland) really unaffected in mutants? If mutant pharyngeal phenotype indeed is restricted to thyroid this should be documented.

Response: Thank you for your valuable suggestions. We measured the overall embryo size (including length, ear and eye) and the results showed that there was no obvious difference between the mutants and their siblings at 5dpf. We have added these results in the manuscript (line 254 and Figs. S2B-S2E).

We observed the development of another two organs, parathyroid and ultimobranchial gland, which derived from pharyngeal endoderm, by WISH. The results showed that compared with siblings the ultimobranchial gland (marked by *calca*) and parathyroid (marked by *gcm2*) were unaffected in 3 dpf and 5 dpf mutants of zebrafish (line 255-259 and Figs. S2F- S2G), which data have been added to the revised manuscript.

4) Morphometric calculations of mouse thyroid shape and size (Fig. 6) infer a role of Hgf/Met pathway prior to E11.5, which represents the end of descensus and start of bilateral elongation of the midline thyroid primordium. What happens thereafter in mutants? Is thyroid phenotype further aggravated with impact on subsequent growth and glandular development, or do you see catch-up of cell proliferation and normal progression the bilobation process? Notably, Fgf10 regulates thyroid developmental growth from E12.5 and onwards (Liang et al Development 145:dev146829, 2018). Thus, it would should be possible to answer whether Hgf/Met action is limited to a previous developmental stage, as suggested by current zebrafish data, and also whether the mutant phenotype merely is the result of delayed thyroid development.

Normal S-T4 and only doubled S-TSH levels postnatally are consistent with subclinical hypothyroidism and argue that thyroid tissue volume and function essentially are sufficient, but is the gland smaller than in wildtype?

Response: Thank you for your valuable suggestions. We furtherly observed the phenotype of thyroid primordium in Met-CKO mice after E11.5. The results have been added at line 361-364 and line 377-379, and discussed the possible developmental mechanism as follows: “Notably, we found the bifurcation of thyroid primordium were significantly delayed in Met-CKO mice at the early or late E11.5 when compared with that in wildtype mice. However, the bilobation process of thyroid in Met-CKO mice

had caught up with that in wildtype mice at E12.5 and E13.5 (Fig. 6Da'-6Dd'). Moreover, the thyroid of Met-CKO mice had normal formation of the left and right lobes at E15.5 (Fig. 6De'-6Df'). These data suggested the HGF/Met pathway play key roles in the bifurcation of thyroid primordium, but did not influenced the bilobation. In fact, thyroid developmental growth from E12.5 and onwards in mice were regulated by Fgf10 pathway, which corresponded to the bilobation during thyroid development (Liang et al Development 145:dev146829, 2018)." (line 424-434)

Minor:

- maybe mutant "families" should be rephrased to mutant strains or lines

Response: Thank you for your advice. We have rephrased "families" to "strains" in the revised manuscript.

- Figs. 1g and h might be excluded or at least reduced in size to conform with other.

Response: Thank you for your suggestion. We have reduced the picture size (Figs.1g - 1h).

- Analysis of TSH (thyrotropin) amounts should preferably be added as supplementary to Figs. 3i-l to estimate level (differences) of hypothyroidism among mutants.

Response: Thank you for valuable suggestion. We have added the analysis of TSH amounts in figures (Figs. 3M-3N and line267-269).

- Legend to Fig. 3M is lacking key to abbreviations (PA and H) and numbers (1-5). Pleass check all legends for similar.

Response: Thank you for your carefully reviewing our paper. We have added the abbreviations in legend (B, brain; p, pharyngeal; PA, pericardial aorta; H, heart; 1-5, cartilage; *, thyroid follicles). And we have checked all legends for similar.

- Morphometric calculation of cell numbers (e.g. in Figs. 3d-h and n) should be

indicated. Both Methods section and figure legend(s) are lacking this information.

Response: we felt sorry for our careless mistakes. The figure legend records "Quantification of the number of thyroid follicles in 1.5-month-old WT and metI284N mutated zebrafish", however, the figure shows "number of thyroid follicular cells per fish". Indeed, the Y-axis should be "number of follicles" rather than "number of mature follicular cells". We have revised them.

- Figs. 5a-j based on cell line analysis does not add much and may be replaced by other more of interest to the paper's major content. The results are largely confirming previous wellknown effects of Hgf/Scatter factor and be removed in toto or switched to Supplementary.

Response: Thank you for your suggestion. We have moved the Figs. 5a-j to Supplementary (Figs. S5).

- Fig. 6A, the IF images are generally of poor quality. Moreover, it is expected that phospho-ERK designating activation of the MAPK pathway should be translocated to the nucleus and captured by IF at least in some cells, which is not the case. This raises doubt on antibody specificity, alternatively on biological effect.

Response: Thank you for your valuable suggestion. We re-performed the immunofluorescence and used the arrowheads to indicate the phospho-ERK (pErk) located in the nucleus of partial thyrocytes (co-expression with Ttf-1) in Fig. 6E. Meanwhile, we also found that the number of the thyrocytes with positive pErk in nucleus in the leading edges of the thyroid primordium at E11.5 significantly reduced in Met-CKO mice when compared with that in wildtype mice (dashed boxes in Fig. 6E). Which all added into the revised manuscript (Line369-376).

- English in need of editing preferably by native.

Response: Thank you for your suggestion. We have carefully examined the paper to the best of our ability and polished the English using Spring Nature Author Services

(<https://authorservices.springernature.com/language-editing/>). These changes did not influence the content and framework of the paper.

Reviewer #3 (Remarks to the Author):

Summary of the Manuscript

The MS of Fang et al. describes defective thyroid gland development in three

zebrafish mutant lines as a result of disruption of HGF-Met signaling. The mutant lines were previously generated by the same group in an ENU mutagenesis screen and were selected for further studies because of a unique thyroid phenotype. The group mapped the mutations to two loci encoding for hgfa and met, two genes that act in the same signaling pathway (hepatocyte growth factor signaling) as hgfa encodes a ligand for the receptor tyrosine kinase met. For both hgfa and met, the Zebrafish Information Network (ZFIN) database lists quite a number of mutant alleles but a thyroid developmental phenotype has so far not been reported. The authors provide limited data on the expression of the two genes showing expression within (met) or adjacent to (hgfa) the developing zebrafish thyroid primordium. A series of additional in vitro experiments confirmed that the met mutations p.I284 and p.E217 impaired met activity. Rescue experiments based on injection of WT met or WT hgfa could rescue the thyroid phenotypes in 5-day-old zebrafish.

The authors next characterized the thyroid phenotypes in some more detail showing that late phases of thyroid development are specifically affected, that defective thyroid morphogenesis results in hypothyroidism and that adult mutant fish have a reduced number of thyroid follicles. Live imaging of transgenic thyroid-specific reporter lines convincingly visualizes the defective expansion of the zebrafish thyroid.

The authors next used a small molecule screen and showed that inhibition of ERK but not PI3K or STAT3 intracellular signaling phenocopies the thyroid defects seen in mutant fish. In a very interesting section of the MS, the authors show that reduced cell adhesion is critical for thyroid expansion in zebrafish and that increased cell adhesion in mutants with perturbed HGF-Met signaling is likely one critical factor causing the thyroid defects.

In the final part of the MS, the authors describe limited phenotyping studies of conditional Met-KO mouse (thyroid-specific invalidation of Met). The data presented appear to show impaired ERK signaling in the thyroid primordium of KO embryos but are not very convincing in the way the data were presented.

General comments

The identification of HGF-Met signaling as a critical pathway to regulate late expansion of the thyroid is a novel finding. To my knowledge, the HGF-Met pathway has not yet been linked to defects in thyroid development and function in mouse models or in the context of congenital hypothyroidism in human. A particular strength of the study by Fang et al. is the demonstration that HGF-Met signaling is linked to Snail-mediated regulation of cell-cell adhesion and that tight regulation of adhesion dynamics could be critical during certain phases of thyroid primordium expansion. I appreciate the attempts by the authors to translate their findings in zebrafish to mouse thyroid organogenesis to demonstrate the broader role of this morphogenetic process but the observations in embryonic mice are poorly presented.

Despite the novelty of the findings and the potential relevance of the reported data to better understand thyroid morphogenesis, the manuscript in its current form has not sufficient quality to recommend it for publication in Nature Communications.

Below, I summarize some of my major concerns.

Major concerns

1)The use of English language needs to be greatly improved. In several instances, a poor English prevented me from understanding what exactly the authors intended to describe. Unfortunately, in its current form, the MS is written for a very small community of readers interested or experienced in thyroid morphogenesis and the MS might fail to attract a broader readership. Due to my personal expertise in pharyngeal organogenesis in zebrafish and mouse embryos, I could follow most descriptions and explanations provided by the authors. However, when reading the MS, I had the strong expression that readers that are not familiar with zebrafish or mouse thyroid organogenesis might experience great difficulties in understanding the anatomical details or developmental

processes described.

The MS would benefit greatly from short introductory notes how thyroid development proceeds (particularly in zebrafish) with some more details on anatomical landmarks that characterize the environment in which the thyroid primordium develops.

Response: Thank you for your suggestions. We have carefully examined the paper to the best of our ability and polished the English using Spring Nature Author Services (<https://authorservices.springernature.com/language-editing/>). These changes did not influence the content and framework of the paper.

We were sorry for the unclear description for the thyroid development proceeds of zebrafish and mouse embryo. We have provided the more details about the thyroid primordium development of zebrafish and mice in the introduction of the revised manuscript (line 86-97).

2) The authors should also improve the use of an appropriate terminology to describe anatomical details and morphogenetic processes. For example, avoid wording like: “dot-like” (line 91), “resembling a ball positioned at the bottom of the aortic arch,” (line 118), “scattering and migration abilities” and “from a rounded ball to an elongated bar that projected bilaterally”(line 373). I also suggest not to use the term “congenital hypothyroidism” (line 211) in relation to zebrafish embryo models. Congenital hypothyroidism describes a condition of reduced circulating TH at birth, so the term is not applicable to zebrafish embryos.

Response: Thank you for your careful review and giving many valuable suggestions. We have corrected all the inappropriate terminology that you listed above in the paper according to the reference (Opitz et al. Dev Biol. 2012;372(2):203-216). And change the term “congenital hypothyroidism” to “hypothyroidism”.

3) I strongly disagree with the description of the main phenotype as a defect in “rostral” elongation. As the authors demonstrate neatly in their live imaging experiments, there is clearly a defect in the “caudal” expansion of the primordium.

The apparent expansion/elongation of the normal zebrafish thyroid primordium between day 2 and day 5 is due to the addition of cells to the caudal portion of the primordium (as the movie nicely show). The many WISH images in Figure 1 also show that the caudal portion of the thyroid is missing.

Response: Thank you for your comments and we felt sorry for our poor English. We have corrected the description of “rostral” by “caudal”.

4) Selected specific comments

I am not satisfied with the quality of some WISH experiments. Particularly the stainings of *hgfa* and also of *foxe1* are very weak and do not show all structures that the express these genes (i.e. there is labeling of the pharyngeal epithelium by the *foxe1* probe). This comment relates to the statements by the authors that *hgfa* expression adjacent to the thyroid was only detectable between 48 and 55 hpf but the defects in thyroid expansion happened during the period from 55 and 120 hpf. This would imply that HGF-Met signaling is active only during the onset of thyroid expansion with lasting consequences even in the absence of continuous *hgfa* expression near the met-expressing thyroid.

Response: Thank you for your valuable comments. We have repeated the WISH experiments for *foxe1* to get much clearly pictures (Fig. S4A). The weak staining of *hgfa* by WISH were caused by the low level expression of *hgfa* in zebrafish embryos. In order to improve the quality of *hgfa* WISH experiments, we used Tg(*tg:GFP*) transgenic embryos to do fluorescence in situ hybridization and immunofluorescence staining to mark *hgfa* and *tg* more clearly. The results have been combined into the Fig. 2 in the revised manuscript. Again, the *hgfa* signal in 72hpf embryos around the thyroid has not been observed by in situ hybridization with fluorescence (data not shown). The consequence of *hgf/met* defects is obvious and we think the reason for this phenotype is either the sensitivity of detection method is limited and maybe there is some *hgf* expression but can not be detected by the method we used or the *hgf/met* signaling is active only during the onset of thyroid expansion, which has been supported by the results of the Erk pathway inhibitors (Fig 4D).

5) Graphs in Figure 3D and 3G visualize the number of “mature follicular cells”. I see many more cells in the corresponding images in 3C and 3F. The same is true for the graph in Figure 3N. Please check.

Response: Thank you for your carefully reviewing our paper and we felt sorry for the carelessness. We mean “the number of thyroid follicles”. We corrected the mistakes (Fig. 3).

6) Line 246 – 248: the authors note “These data indicated that the Hgfa-met pathway probably regulated rostral elongation along a pair of hypobranchial arteries in zebrafish during late thyroid development”. It would be very informative to visualize the relative positioning of the thyroid and the paired hypobranchial arteries in WT and mutant fish during the process of elongation. A live imaging video and some images in Figure S3B are provided but only for the period before elongation. With respect to the specific anatomical relationship between the pharyngeal vessels and the thyroid the question arises if it is the vessels that express hgfa. From a larger conceptual perspective, this is an extremely relevant question because if vessels are the source of hgfa, then HGF-Met signaling could also be compromised in cases of vascular maldevelopment.

Response: Thank you for your valuable comments. We have provided the images for the period of thyroid elongation of zebrafish (Fig. S4B). The results showed that truncated *hgfa* mutation in zebrafish does not affect the development of vessels surrounding the thyroid at 72 hpf embryos (Fig. S4B). The data have been added in the revised manuscript.

7) Figure 6. I could not orient myself on the images shown in Figure 6. Are these really sagittal sections? How can a “bifurcation index” be calculated on sagittal sections? Please provide additional images that contain anatomical landmarks (such as the nearby vessels).

Response: Thank you for your valuable comments. We were sorry for the misleading

statement. The pictures in Fig 6 are transverse sections instead of sagittal sections. We have provided new figures by the transverse sections of thyroid primordium in E11.5 mice embryo (Fig. 6A) and marked the relative position of the thyroid primordium (th) to the surrounding vessels (the third pharyngeal arch artery, aortic arch). We also observed the thyroid development in the mice embryo at E12.5 to E15.5. We found that HGF/Met pathway play key roles in the bifurcation of thyroid primordium, but did not influence the bilobation. These data have been added in the revised manuscript (line 363-364 and Fig.6D).

REVIEWER COMMENTS

Reviewer #1 (Remarks to the Author):

The authors have addressed all the concerns raised by me. I recommend the manuscript for publication.

Reviewer #2 (Remarks to the Author):

This reviewer has no further comments or suggestions to authors.

Reviewer #3 (Remarks to the Author):

The review of the revised manuscript of Fang et al. was challenging to me. I really think that the core findings of the study are novel and exciting. The hgf/met pair identification is an important example demonstrating the role of a precise stromal-derived factor and its cognate thyrocyte receptor in early thyroid morphogenesis. Thyroid follicles are epithelial structures and the described role of hgf/met in regulating an EMT-mediated process critical for formation of new naïve follicles is very interesting from the conceptual perspective. Moreover, some of the new data presented by the authors (Fig. 2A_a",b") suggest that pharyngeal vessels are a likely source of hgfa (as judged from the shape of the hgfa expression domain). If confirmed, hgfa would constitute a first candidate molecule that contributes to the so often "anticipated" vascular guidance role during early thyroid morphogenesis. The importance of identifying one specific vascular-born factor that affects morphogenic processes can not be overstated.

Unfortunately, the quality of reporting is in to many places not satisfying. The English still needs a lot of editing. The methods need more care for details.

Specific comments:

In Fig. 2H; "Western blots analyzed the expression after treated with or without human HGF α protein". No HGF α conditions shown in this panel.

Fig. 3O shows thyroid follicles in the hypobranchial region in 1.5 months old fish. TH levels indicate a hypothyroid condition but the authors did not mention any change in follicular morphology. There are numerous zebrafish studies that showed a classical goitrous thyroid phenotype in experimentally induced or genetically determined hypothyroidism. Was there any change in thyroid follicular cell morphology in these hypothyroid mutants (given the proposed increase in TSH; see comments below).

Fig. 2A, 3C,D, 4A and 5A-B. the figure legends should include information on the orientation of the tissue sections shown (e.g., horizontal or sagittal sections, anterior is to the right)

Terminology issues:

The Tübingen background would be the STRAIN of zebrafish that was used for mutagenesis. The mutation procedure would generate mutant LINES. Please use the term LINE throughout the manuscript.

The terms EMBRYO (up to 72 hpf) and LARVAE are sometimes used for the same group of

fish in a single sentence. By convention, the zebrafish “embryo” takes the name of “larvae” from 72 hpf onwards.

Change “continuously activated MEK2” to “constitutively active MEK”

From the perspective of reproducibility, the methods section still lacks in many cases minimal information on procedures, validation efforts or how often specific analyses had been replicated

A few examples on incomplete information in the methods section:

It is a common standard to clearly state the origin of each transgenic zebrafish lines (reference to original publication describing the generation of the line is a standard in the field).

The authors generated a new mouse lines (floxed Met allele). Apart from mentioning primer pairs, no information on genotyping results is presented. These mice were crossed with Pax8-Cre to generate a thyroid-specific KO model, again no information at all about recombination efficiency undermining any conclusion about the mild thyroid phenotype.

The description of the zebrafish constructs reads awkwardly. The use of Gateway strategy is worth mentioning to understand the construction process. Experiments using pTol2 construct injections would benefit from a bit more detail on injection procedure (e.g., amount of plasmid).

The method used to estimate TSH levels in zebrafish by means of aqueous extracts of whole zebrafish tissue is completely new to me and I am actually surprised that this approach could work for a peptide hormone. How did the authors validate that the human?/mouse? kits for TSH can indeed detect zebrafish TSH. The authors referred to a paper by Yung 2011 but this paper does not describe any TSH measurements.

Counting of thyroid follicles in adult fish. Images in Fig. 3O show sagittal sections (not horizontal as described in methods). How was the counting done (manually section by section?). How many sections per fish were analyzed to derive a total number per fish? How many fish were analyzed?

Western blotting. No information apart from incomplete description of antibodies that were used.

Response to the reviewers point by point:

REVIEWER COMMENTS

Reviewer #3 (Remarks to the Author):

The review of the revised manuscript of Fang et al. was challenging to me. I really think that the core findings of the study are novel and exciting. The hgf/met pair identification is an important example demonstrating the role of a precise stromal-derived factor and its cognate thyrocyte receptor in early thyroid morphogenesis. Thyroid follicles are epithelial structures and the described role of hgf/met in regulating an EMT-mediated process critical for formation of new naïve follicles is very interesting from the conceptual perspective. Moreover, some of the new data presented by the authors (Fig. 2A_a””,b””) suggest that pharyngeal vessels are a likely source of hgfa (as judged from the shape of the hgfa expression domain). If confirmed, hgfa would constitute a first candidate molecule that contributes to the so often “anticipated” vascular guidance role during early thyroid morphogenesis. The importance of identifying one specific vascular-born factor that affects morphogenic processes can not be overstated.

Response:

Bifurcation during the thyroid development is the transverse elongation of the thyroid primordium along the third pharyngeal arch arteries after its descent (Fagman et al., 2004; Fagman et al., 2007; Kameda et al., 2009; Nilsson and Fagman, 2017), however, the factors responsible for this process remain unclear. A previous study revealed that sonic hedgehog (Shh) signals regulate the bifurcation, which is likely secondary to severe malformation of the vascular tree emerging from the outflow tract (Zhang et al., 2005). Thus, the reviewer’s hypothesis that **“hgfa derived from vascular as a first candidate molecule to contribute to the ‘anticipated’ vascular guidance role during the bifurcation during the thyroid development”** is very interesting and meaningful. In fact, hgfa has been reported to be expressed in the endothelial cells (Leung et al., *Oncogene*. 2017). However, we further analyzed the single-cell RNA-seq data of mouse thyroid (Yang et al., *Nat Commun*. 2023) in our lab, and found that fibroblasts were the main source of *Hgf* in thyroid tissue and that there was almost no *Hgf* expression in endothelial cells (with *cdh5* as a specific marker of endothelial cells). These data suggested that the expression of hgfa was more abundant in fibroblasts than that in endothelial cells in thyroid tissue. We thus presumed that the *hgfa* derived from the fibroblasts, rather than from endothelial cells of the surrounding vasculature in thyroid tissues, might play a key role in the bifurcation during thyroid development. However, additional specific experiments, such as conditional knockout of *hgfa* in distinct cells are needed in the future, to clarify whether the effect of *hgfa* on bifurcation during thyroid development was derived from endothelial cells of the vasculature surrounding the thyroid gland or from the fibroblasts in the thyroid stroma. We have

added the *Hgf* expression analysis using the single cell RNA-seq data of mouse thyroid as Figs.S2.

Figures showed the expression of *Hgf* (*hgfa* in zebrafish) in mouse thyroid was mainly in the fibroblasts. *Cdh5* was a specific marker for endothelia cells. *Pdgfra* was a specific marker for fibroblasts. *Hgf* was mainly expressed in the fibroblasts, followed by myeloid cells. There was almost no *Hgf* expression of in endothelia cells.

Unfortunately, the quality of reporting is in to many places not satisfying. The English still needs a lot of editing. The methods need more care for details.

Response: Thank you for your comments and suggestions. We have modified the manuscript accordingly and polished the English by Springer Nature author Services. The detailed corrections are listed below point by point. Revised portion are marked in blue in the paper.

1 Specific comments:

1) In Fig. 2H; “Western blots analyzed the expression after treated with or without human HGF α protein”. No HGF α conditions shown in this panel.

Response: Thank you for your carefully reviewing our paper. We are sorry for the careless mistake. We have revised the figure of Fig.2H in the revised manuscript.

2) Fig. 3O shows thyroid follicles in the hypobranchial region in 1.5 months old fish. TH levels indicate a hypothyroid condition but the authors did not mention any change in follicular morphology. There are numerous zebrafish studies that showed a classical goitrous thyroid phenotype in experimentally induced or genetically determined hypothyroidism. Was there any change in thyroid follicular cell morphology in these hypothyroid mutants (given the proposed increase in TSH; see comments below).

Response: In our previous manuscript, we have found that the number of thyroid follicles was fewer in the 1.5 mpf zebrafish mutants than in the wild-type siblings of the met^{I284N} line, as shown by H&E staining of sagittal sections (Figs. 3O-3Q). Notably, we also found that the distribution of thyroid follicles in mutants in the met^{I284N} line was also different from that in the WT siblings, which mainly gathered in the anterior position (Fig. 3O). Moreover, some large follicles have also been observed in the thyroid of mutant zebrafish (as shown in the Fig. S3H). The morphologic characterization of the thyroid follicles in the 1.5 mpf zebrafish mutants was similar with the phenotype of human classical goitrous, indicated that the hypothyroidism caused by deficiency of HGF/Met have partially been compensated by hyperplasia of the thyroid follicles after birth of the mutant zebrafish. **These data have been added in the results of the revised manuscript as Fig. S3H.**

3) Fig. 2A, 3C,D, 4A and 5A-B. the figure legends should include information on the orientation of the tissue sections shown (e.g., horizontal or sagittal sections, anterior is to the right)

Response: Thank you for your carefully reviewing our paper. We have added information on the orientation of the tissue sections shown in the figure legends.

2 Terminology issues:

1) The Tübingen background would be the STRAIN of zebrafish that was used for mutagenesis. The mutation procedure would generate mutant LINES. Please use the term LINE throughout the manuscript.

Response: Thank you for your advice. We have revised the description in our revised manuscript.

2) The terms EMBRYO (up to 72 hpf) and LARVAE are sometimes used for the same group of fish in a single sentence. By convention, the zebrafish “embryo”

takes the name of “larvae” from 72 hpf onwards.

Response: Thank you for your valuable advice. We have revised the description in our revised manuscript.

3) Change “continuously activated MEK2” to “constitutively active MEK”

Response: Thank you for your suggestion. We have changed “continuously activated MEK2” to “constitutively active MEK” in the revised manuscript.

3 From the perspective of reproducibility, the methods section still lacks in many cases minimal information on procedures, validation efforts or how often specific analyses had been replicated.

Response: We are sorry for the failure to describe the experimental procedure in detail in the methods section, and we have added experimental details in our revised manuscript.

1) A few examples on incomplete information in the methods section:

It is a common standard to clearly state the origin of each transgenic zebrafish lines (reference to original publication describing the generation of the line is a standard in the field).

Response: We have stated the origin of each transgenic zebrafish lines in the revised manuscript.

2) The authors generated a new mouse lines (floxed Met allele). Apart from mentioning primer pairs, no information on genotyping results is presented. These mice were crossed with Pax8-Cre to generate a thyroid-specific KO model, again no information at all about recombination efficiency undermining any conclusion about the mild thyroid phenotype.

Response: Thank you for your valuable suggestion. We have added the information on genotyping results and recombination efficiency of the Met-CKO mouse model in the manuscript (Fig.S7).

3) The description of the zebrafish constructs reads awkwardly. The use of Gateway strategy is worth mentioning to understand the construction process. Experiments using pTol2 construct injections would benefit from a bit more detail on injection procedure (e.g., amount of plasmid).

Response: We have revised the description of the zebrafish constructs and added the details on injection procedure in our revised manuscript according to the suggestion of reviewer 3# .

4) The method used to estimate TSH levels in zebrafish by means of aqueous extracts of whole zebrafish tissue is completely new to me and I am actually surprised that this approach could work for a peptide hormone. How did the authors validate that the human?/mouse? kits for TSH can indeed detect zebrafish TSH. The authors referred to a paper by Yung 2011 but this paper does not describe any TSH measurements.

Response: Thank you for your comments. We used the zebrafish specific TSH kit (CAMILO, Nanjing, China, Product No. 2F-KMLJf91039) to detect TSH level of zebrafish (Wei et al., *Chemosphere*. 2022), and the mouse TSH kit (Cloud Clone Corp, China, Product No. CEA463Mu) to detect mouse TSH levels. In the two kits, the antibodies respectively targeted to zebrafish or mouse TSH have been used. We have revised the description in the revised manuscript.

5) Counting of thyroid follicles in adult fish. Images in Fig. 3O show sagittal sections (not horizontal as described in methods). How was the counting done (manually section by section?). How many sections per fish were analyzed to derive a total number per fish? How many fish were analyzed?

Response: For each group, three fish were sacrificed to evaluate the gross morphology of the thyroid. All sections with follicles were taken to count the number, manually section by section. In this experiment, 19-23 sections per wild-type fish and 10-17 sections per mutant fish were analyzed. We have added the details in our manuscript.

6) Western blotting. No information apart from incomplete description of antibodies that were used.

Response: We have added the details of western blotting in the Methods section. For western blotting, protein samples were separated on 10% SDS-PAGE and transferred onto nitrocellulose membranes. These were blocked with 5% FBS (SigmaAldrich) in 0.1% TBST for 2 h at room temperature, and then incubated overnight at 4°C with primary antibody, washed 3 times with TBST, and incubated for 2 h at room temperature with HRP-conjugated secondary antibody (1:5000). Following 3×TBST washes the membranes were developed with Immunoblot Forte Western HRP (Millipore), according to the manufacturer's instructions, and imaged using an Amersham™ Imager 600 (GE Healthcare).

Reference:

1. Fagman, H., Grände, M., Gritli-Linde, A., and Nilsson, M. (2004). Genetic deletion of sonic hedgehog causes hemiagenesis and ectopic development of the thyroid in mouse. *The American journal of pathology* 164, 1865-1872.
2. Fagman, H., Liao, J., Westerlund, J., Andersson, L., Morrow, B.E., and Nilsson, M. (2007). The 22q11 deletion syndrome candidate gene *Tbx1* determines thyroid size and

positioning. *Hum Mol Genet* 16, 276-285.

3. Kameda, Y., Ito, M., Nishimaki, T., and Gotoh, N. (2009). FRS2alpha is required for the separation, migration, and survival of pharyngeal-endoderm derived organs including thyroid, ultimobranchial body, parathyroid, and thymus. *Dev Dyn* 238, 503-513.
4. Nilsson, M., and Fagman, H. (2017). Development of the thyroid gland. *Development* 144, 2123-2140.
5. Zhang, Z., Cerrato, F., Xu, H., Vitelli, F., Morishima, M., Vincentz, J., Furuta, Y., Ma, L., Martin, J.F., Baldini, A., et al. (2005). Tbx1 expression in pharyngeal epithelia is necessary for pharyngeal arch artery development. *Development* 132, 5307-5315.
6. Leung, E., Xue, A., Wang, Y., Rougerie, P., Sharma, V. P., Eddy, R., Cox, D., & Condeelis, J. (2017). Blood vessel endothelium-directed tumor cell streaming in breast tumors requires the HGF/C-Met signaling pathway. *Oncogene*, 36, 2680–2692.
7. Yang R.M., Song, S.Y., Wu, F.Y., Yang, R.F., Shen, Y.T., Tu, P.H., Wang, Z., Zhang, J.X., Cheng, F., Gao, G.Q., Liang, J., Guo, M.M., Yang, L., Zhou, Y., Zhao, S.X., Zhan, M., Song, H.D. Myeloid cells interact with a subset of thyrocytes to promote their migration and follicle formation through NF- κ B. (2023). *Nat Commun* 14, 8082.
8. Wei G., Zhang CX., Jing Y., Chen X., Song HD., Yang L. (2022). The influence of sunitinib and sorafenib, two tyrosine kinase inhibitors, on development and thyroid system in zebrafish larvae. *Chemosphere* 308, 136354.

REVIEWERS' COMMENTS

Reviewer #3 (Remarks to the Author):

The recent revision by the authors greatly improved the quality of the manuscript. From my perspective, it now meets the standards for publication in NatComm. I would like to congratulate the authors for their meticulous work. When reviewing the recent manuscript version, I only found some minor points that need to be addressed (outlined below).

Minor comments

Main text:

Line 97 typo, nkx2.4b

Methods:

please add information to the methods section how the EdU labeling experiment was done. To understand the values that you obtained, it is critical to know the length of the pulse and a possible chase period.

Figures:

Figure 1 D: Please check the lower panel of Sanger sequencing results. All three graphs for “siblings” show hets, so labelling the group of panels as heterozygous siblings would be appropriate. The base sequences shown below each sequencing profile are not consistent. The left panel mentions A but the sequence profile shows a het with A/T, the middle panel mentions C but the sequence profile shows a het with C/T, and the right panel mentions A but the sequence profile shows a het with A/T.

Figure 3: Use capital letter O for labelling of panel O.

Figure 6: there is a mix of labels to annotate the embryonic age. Please use one consistent label; either E for embryonic day or dpc for days post-conception

Supplemental Figures:

Figure S3: Please provide a value for the length of the scale bar in Fig. S3H

Figure S4: Line 25 Typo “thyrocyte(s) proliferation”, please add information that “anterior is to the right” in Fig. S4; recommend to change Edu to EdU in Figure panels;

Response to the reviewers point by point:

REVIEWER COMMENTS

Reviewer #3 (Remarks to the Author):

The recent revision by the authors greatly improved the quality of the manuscript. From my perspective, it now meets the standards for publication in NatComm. I would like to congratulate the authors for their meticulous work. When reviewing the recent manuscript version, I only found some minor points that need to be addressed (outlined below).

Minor comments

Main text:

Line 97 typo, nkx2.4b

Response: Thank you for your carefully reviewing our paper. We have revised the typo in our manuscript.

Methods:

please add information to the methods section how the EdU labeling experiment was done. To understand the values that you obtained, it is critical to know the length of the pulse and a possible chase period.

Response: Thank you for your valuable suggestion. We have added the information to the methods section how the EdU labeling experiment was done (Line 722-725).

Figures:

Figure 1 D: Please check the lower panel of Sanger sequencing results. All three graphs for “siblings” show hets, so labelling the group of panels as heterozygous siblings would be appropriate. The base sequences shown below each sequencing profile are not consistent. The left panel mentions A but the sequence profile shows a het with A/T, the middle panel mentions C but the sequence profile shows a het with C/T, and the right panel mentions A but the sequence profile shows a het with A/T.

Response: Thank you for your valuable suggestion. We have revised the group of panels as heterozygous siblings and revised the base sequences.

Figure 3: Use capital letter O for labelling of panel O.

Response: Thank you for your suggestion. We have used capital letter O for labelling of panel O.

Figure 6: there is a mix of labels to annotate the embryonic age. Please use one consistent label; either E for embryonic day or dpc for days post-conception

Response: Thank you for your carefully reviewing our paper. We have revised the labels and used consistent label to annotate the embryonic age in Figure 6.

Supplemental Figures:

Figure S3: Please provide a value for the length of the scale bar in Fig. S3H

Response: Thank you for your carefully reviewing our paper. We have added the value for the length of the scale bar in Fig. S3H.

Figure S4: Line 25 Typo “thyrocyte(s) proliferation”, please add information that “anterior is to the right” in Fig. S4; recommend to change Edu to EdU in Figure panels;

Response: Thank you for your suggestion. We have revised typo “thyrocytes proliferation” and added information that “anterior is to the right” in Fig. S4; we have changed Edu to EdU in Figure panels.